# Zfp281 is essential for mouse epiblast maturation through transcriptional and epigenetic control of Nodal signaling

Xin Huang[1,2†], Sophie Balmer[3†], Fan Yang[1,2,4], Miguel Fidalgo[1,2,5], Dan Li[1,2,6], Diana Guallar[1,2], Anna-Katerina Hadjantonakis[3*], Jianlong Wang[1,2,6*]

[1]The Black Family Stem Cell Institute, Icahn School of Medicine at Mount Sinai, New York, United States; [2]Department of Cell, Developmental and Regenerative Biology, Icahn School of Medicine at Mount Sinai, New York, United States; [3]Developmental Biology Program, Sloan Kettering Institute, Memorial Sloan Kettering Cancer Center, New York, United States; [4]Department of Animal Biotechnology, College of Veterinary Medicine, Northwest A&F University, Yangling, China; [5]Departamento de Fisioloxia, Centro de Investigacion en Medicina Molecular e Enfermidades Cronicas, Universidade de Santiago de Compostela, Santiago, Spain; [6]The Graduate School of Biomedical Sciences, Icahn School of Medicine at Mount Sinai, New York, United States

**\*For correspondence:**
hadj@mskcc.org (A-KH);
jianlong.wang@mssm.edu (JW)

[†]These authors contributed equally to this work

**Competing interests:** The authors declare that no competing interests exist.

**Abstract** Pluripotency is defined by a cell's potential to differentiate into any somatic cell type. How pluripotency is transited during embryo implantation, followed by cell lineage specification and establishment of the basic body plan, is poorly understood. Here we report the transcription factor Zfp281 functions in the exit from naive pluripotency occurring coincident with pre-to-post-implantation mouse embryonic development. By characterizing *Zfp281* mutant phenotypes and identifying Zfp281 gene targets and protein partners in developing embryos and cultured pluripotent stem cells, we establish critical roles for Zfp281 in activating components of the Nodal signaling pathway and lineage-specific genes. Mechanistically, Zfp281 cooperates with histone acetylation and methylation complexes at target gene enhancers and promoters to exert transcriptional activation and repression, as well as epigenetic control of epiblast maturation leading up to anterior-posterior axis specification. Our study provides a comprehensive molecular model for understanding pluripotent state progressions in vivo during mammalian embryonic development.
DOI: https://doi.org/10.7554/eLife.33333.001

## Introduction

Development of an organism from a fertilized egg involves the coordination of cell lineage specification coupled with the establishment of cardinal axes (including the anterior-posterior (A-P) axis), to build a blueprint for the body plan (*Arnold and Robertson, 2009*). The earliest stages of mammalian development culminate in the formation of a blastocyst comprising three cell lineages: the pluripotent epiblast (Epi) which gives rise to somatic and germ cells, and two extra-embryonic lineages, the primitive endoderm (PrE) and trophectoderm (TE) (*Schrode et al., 2013*). Mouse embryonic stem cells (ESCs) are derived from, and represent an in vitro self-renewing counterpart of, the so-called 'naive' pluripotent epiblast cells of the blastocyst (*Nichols and Smith, 2009*; *Boroviak et al., 2014*). Pluripotency comprises a continuum of states sequentially encompassing naive, formative and ultimately primed pluripotency (*Smith, 2017*). Upon blastocyst implantation into the maternal uterus, epiblast cells acquire characteristics of the more developmentally advanced formative and then

primed states of pluripotency, which are represented by epiblast-like cells (EpiLCs) (*Hayashi et al., 2011*) and epiblast stem cells (EpiSCs) (*Brons et al., 2007*; *Tesar et al., 2007*; *Kojima et al., 2014*), respectively. Exploiting the ability of ESCs to transition into EpiLCs/EpiSCs has advanced the mechanistic understanding of epiblast maturation. However, while in vitro studies are instrumental, it remains critical to confirm whether the insights they provide are relevant to events taking place in vivo where the only *bona fide* pluripotent population, the epiblast, normally exists. The transition to more developmentally advanced states of pluripotency occurs in the mouse embryo coincident with its implantation into the maternal uterus, at around embryonic day (E)4.5–5.0, and is marked by several stereotypical morphological and molecular changes (*Bedzhov and Zernicka-Goetz, 2014*). Acquisition of the primed state of pluripotency precedes the onset of gastrulation (at E6.25), the process in which the three embryonic germ layers - ectoderm, mesoderm, and endoderm - are formed (*Tam et al., 2006*; *Arnold and Robertson, 2009*; *Takaoka and Hamada, 2012*).

Key events of epiblast maturation include coordinated expression of specific transcription factors (TFs) across developmental stages. Nanog, Klf4, and Rex1 (also named Zfp42) are highly expressed in the epiblast of the blastocyst and ESCs, whereas Fgf5, Oct6 (also named Pou3f1), and Otx2 are upregulated in the epiblast following embryo implantation, or when ESCs differentiate toward EpiSCs. Factors such as Eomes or T (also named Brachyury) are expressed in gastrulating embryos at the primitive streak (the site where pluripotent cells undergo lineage differentiation) and in EpiSCs. Other pluripotency-associated TFs, such as Oct4, Sox2, and Zfp281, are expressed in the pluripotent epiblast throughout these state transitions, suggesting they may play distinct roles in different pluripotent states, or enable transitions between them. Specific DNA modifications and reorganization of enhancer landscapes also occur during the naive-to-primed transition, together with genome-wide relocation of Oct4, as well as elevated binding of Otx2 and the P300 histone acetyltransferase at enhancers of genes specific to the primed state (*Buecker et al., 2014*; *Yang et al., 2014*). Concomitantly, during early post-implantation embryo development, the A-P axis is established. A-P patterning is not readily recapitulated in ESC or EpiSC cultures since it necessitates cross-talk between the epiblast and its adjacent extra-embryonic tissue, the visceral endoderm (VE) (*Shen, 2007*). In the mouse embryo, distal visceral endoderm (DVE) cells, specified at the late blastocyst stage as a sub-population of the PrE, are critical for A-P axis establishment (*Beddington and Robertson, 1999*; *Takaoka and Hamada, 2012*). At E5.5, DVE cells are localized at the distal tip of the embryo from where they migrate proximally towards the extra-embryonic/embryonic boundary, recruiting a second population (the anterior visceral endoderm or AVE) and defining an anterior to the embryo, thereby establishing an A-P axis (*Stower and Srinivas, 2014*). The TGF-beta ligand Nodal, which is expressed by the epiblast, and several of its pathway components, such as the left-right determination factors (Lefty1 and 2) (*Brennan et al., 2001*), Cripto (also named Tdgf1) (*Ding et al., 1998*), and Foxh1 (*Yamamoto et al., 2001*) are required for DVE/AVE specification, migration, and A-P axis formation (*Brons et al., 2007*; *Takaoka and Hamada, 2012*). Whether epiblast maturation and A-P axis specification can be mechanistically linked remains an open question.

Zfp281 was recently identified as a TF required for the commitment of ESCs to differentiation in culture (*Betschinger et al., 2013*; *Fidalgo et al., 2016*). In this study, we investigate pluripotent state transitions in vivo in their native context, and identify a key role for Zfp281 in early mammalian development. Mouse embryos lacking Zfp281 reach the blastocyst stage and establish a pluripotent epiblast lineage. However, they exhibit defects in epiblast maturation, indicated by the failure to robustly activate Nodal signaling and genes associated with the primed pluripotent state. Hence, they are unable to exit the naive pluripotent state, resulting in a failure to establish an A-P axis. Mechanistically, we demonstrate that Zfp281 functions specifically within the epiblast to coordinate the epigenetic regulators acting to initiate expression of lineage-specific genes and modulate the Nodal signaling pathway.

## Results

### Zfp281 is expressed in early mouse embryos and required for early post-implantation development

To begin to investigate the role of Zfp281 in vivo during mouse embryonic development when the pluripotent epiblast population is established and matures, we determined the localization of the

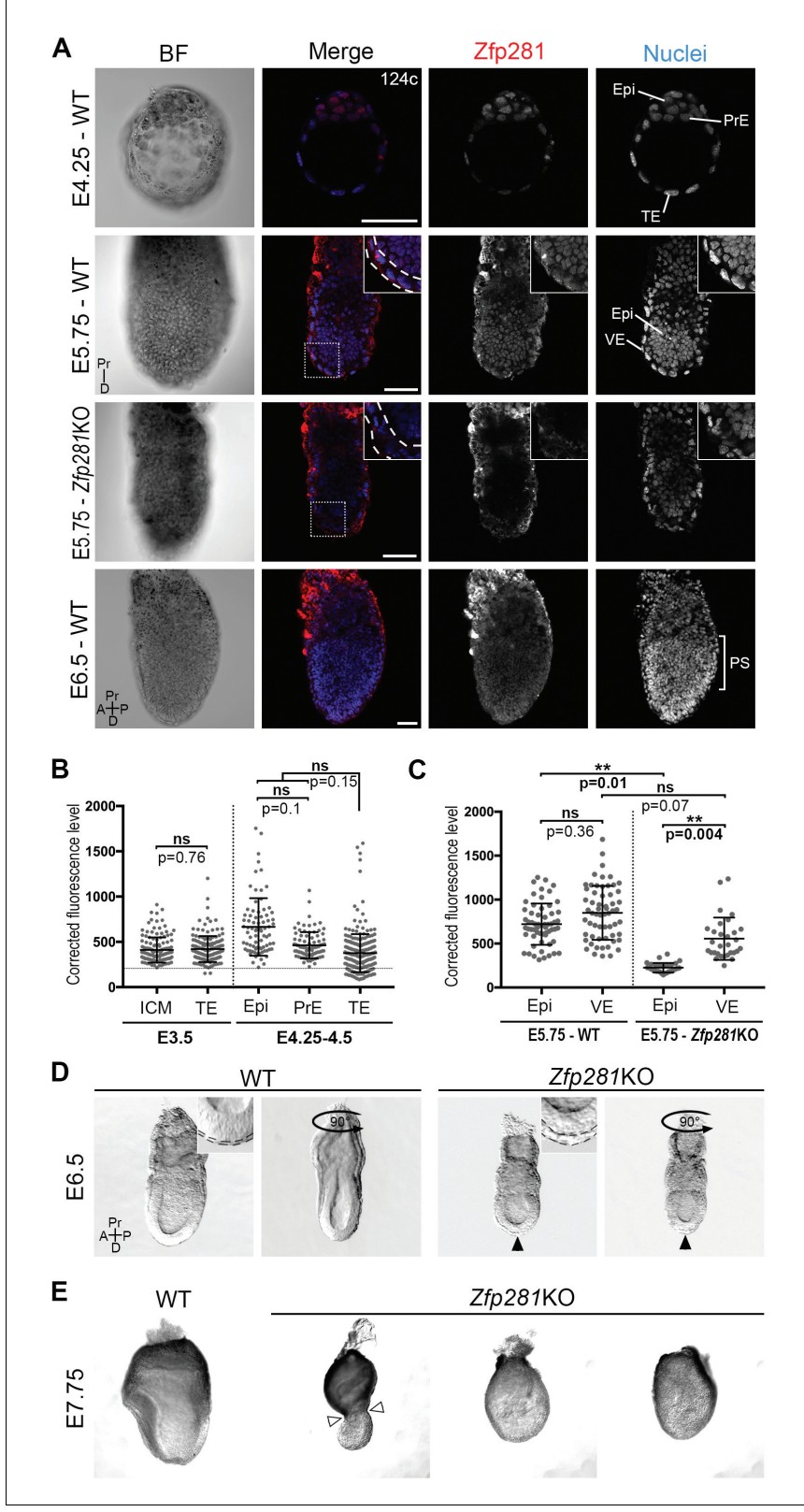

**Figure 1.** Zfp281 is expressed in the pluripotent epiblast and is required for early post-implantation embryo development. (**A**) Single optical sections depicting Zfp281 protein expression in pre- and post-implantation. At pre-implantation, nuclear-localized Zfp281 protein is observed in epiblast (Epi), primitive endoderm (PrE) and trophectoderm (TE) cells, as quantified in (**B**). Zfp281 expression is specific to epiblast at post-implantation (E5.75),
*Figure 1 continued on next page*

*Figure 1 continued*

quantified in (**B**). Immunohistochemistry of *Zfp281*KO embryo at E5.75 shows that the protein is not expressed, confirming the mutant as a protein null. It also reveals VE-specific background. High-magnification insets (top-right) show protein distribution in regions highlighted, white dashed lines delimit the VE layer. At onset of gastrulation (E6.5), Zfp281 is expressed in all epiblast-derived cells. (**B**) Quantification of nuclear levels of Zfp281 using MINS software at mid- (E3.5) and late (E4.25–4.5) blastocyst stage, revealing protein expression in all three cell types. n = 5 embryos (308 cells) for mid-blastocyst stage and three embryos (448 cells) for late blastocyst stage. See *Figure 1—figure supplement 1* for immunohistochemistry of additional stages. (**C**) Quantification of nuclear levels of Zfp281 in VE and Epi cells in WT and *Zfp281*KO embryos at E5.75 using Imaris software. n = 6 embryos for the WT (20 cells per genotype) and n = 3 embryos for *Zfp281*KO. The expression of Zfp281 in the VE was ruled out through quantitative fluorescence level comparisons of wild-type (WT) and *Zfp281*KO embryos, which lack Zfp281 protein. (**D**) At E6.5, the A-P axis is established and WT embryos initiate gastrulation at their posterior, while *Zfp281*KO embryos display a thickened visceral endoderm epithelium (black arrowhead) and no A-P polarity. Insets (top-right) depict thickened VE layer (delineated by black dashed lines) in *Zfp281*KO embryo compared to the WT. (**E**) *Zfp281*KO embryos die around E8.0 and exhibit aberrant gross morphology at E7.75 when compared to WT with either cells of the epiblast layer undergoing apoptosis and/or constriction at the embryonic/extra-embryonic junction (white arrowheads). A = Anterior, p=Posterior, Pr = proximal, D = Distal, BF = brightfield, Scale bars represent 50 µm. Statistical significance was calculated on the average level of corrected fluorescence per embryo using Student T-test.

DOI: https://doi.org/10.7554/eLife.33333.002

The following figure supplements are available for figure 1:

**Figure supplement 1.** Expression and function of Zfp281 in early post-implantation embryo development.
DOI: https://doi.org/10.7554/eLife.33333.003
**Figure supplement 2.** Summary of embryos recovered from intercrosses of heterozygous mice.
DOI: https://doi.org/10.7554/eLife.33333.004
**Figure supplement 3.** *Zfp281*KO phenotype at early post-implantation stages.
DOI: https://doi.org/10.7554/eLife.33333.005

Zfp281 protein by immunohistochemistry (*Figure 1A–C*, *Figure 1—figure supplement 1A*). At E3.5, representing the mid-blastocyst stage, Zfp281 was nuclear-localized and detected at low levels throughout the inner cell mass (ICM) and trophectoderm (TE) (*Figure 1B*, *Figure 1—figure supplement 1A*). This widespread expression was maintained until the late blastocyst stage (E4.5) in both ICM derivatives including Epi and PrE, as well as the TE (*Figure 1A–B*, *Figure 1—figure supplement 1A*). Single-cell quantitative immunofluorescence (*Lou et al., 2014*; *Saiz et al., 2016*) and single-cell microarray data (*Ohnishi et al., 2014*) revealed that Zfp281 expression was elevated in the epiblast of late stage blastocysts (*Figure 1B*, *Figure 1—figure supplement 1B*). At later (post-implantation) stages, Zfp281 expression was restricted to the epiblast and its derivatives (*Figure 1A*) and absent from the visceral endoderm (VE), a squamous epithelium derived from the PrE, which encapsulates the epiblast (*Figure 1C*, *Figure 1—figure supplement 1C*).

Despite its critical function in maintaining ESC pluripotency (*Wang et al., 2008*; *Fidalgo et al., 2011*) and promoting the transition from the naive-to-primed state of pluripotency in vitro (*Fidalgo et al., 2016*), whether Zfp281 plays a role within the epiblast lineage of the developing embryo remained an open question. A conventional knockout (KO) allele of Zfp281 generated using a gene targeting approach (*Fidalgo et al., 2011*) was used for embryo analysis. No homozygous mutant mice were recovered at birth from intercrosses of heterozygous animals, demonstrating a requirement for Zfp281 in embryonic development (*Figure 1—figure supplement 2A*). The analysis of staged embryos revealed that mutants died around E8.0 (*Figure 1—figure supplement 2B*). At E5.5, *Zfp281*KO embryos were recovered at Mendelian ratios and were indistinguishable from their WT and heterozygous littermates by gross morphology (*Figure 1—figure supplement 3A*). However, by E6.0–6.5, *Zfp281* mutants became readily distinguishable from their WT littermates by their smaller size and distinct morphology, exhibiting a thickened VE layer (*Figure 1D*, insets, *Figure 1—figure supplement 3B*, insets with yellow bars). These results suggest a requirement of Zfp281 for both the VE and epiblast, which are likely to be non-cell-autonomous and cell-autonomous, respectively, given the epiblast-specific expression of Zfp281.

To confirm the epiblast-specific function of Zfp281, we produced embryos in which Zfp281 was specifically absent in the epiblast, but in which TE and PrE derivatives were wild-type (WT). To do

this we generated tetraploid (4n) WT <->Zfp281KO ESC chimeras and analyzed them at early post-implantation stages (*Figure 1—figure supplement 3C*). Epiblast-specific loss of Zfp281 produced embryos with a comparable phenotype to that of constitutive gene ablation, with both types of mutant embryos exhibiting a thickened VE (*Figure 1—figure supplement 3D*). The defect observed in tetraploid chimeras comprising *Zfp281*KO ESCs could be partially rescued when a Zfp281 cDNA was transfected into *Zfp281*KO ESCs (referred to as *Zfp281*KO + Zfp281 cDNA), further confirming the epiblast tissue specificity and Zfp281 gene specificity in the observed mutant phenotypes.

By E7.75, mutant embryos appeared grossly abnormal and displayed either a constriction at the embryonic and extra-embryonic boundary (*Figure 1E*, white arrowheads), resembling the DVE/AVE defects observed in mutant embryos lacking Eomes (*Nowotschin et al., 2013*), Otx2 (*Ang et al., 1996*), Lim1/Lhx1 (*Shawlot et al., 1998*; *Costello et al., 2015*), Cripto (*Ding et al., 1998*), Foxh1 (*Yamamoto et al., 2001*), and Foxa2 (*Ang and Rossant, 1994*), or they exhibited extensive cell death and a massive loss of epiblast cells (*Figure 1E*). Together, these results suggest a cell-autonomous requirement for Zfp281 within the epiblast, and a non-cell-autonomous requirement within the VE.

## Deregulation of Nodal signaling and A-P axis specification related genes in *Zfp281*KO embryos

To identify the molecular changes associated with loss of Zfp281, we characterized the transcriptomic differences between WT and *Zfp281*KO embryos. We performed RNA sequencing (RNA-seq) on individual E6.5 embryos, corresponding to the earliest stage at which mutants could be morphologically distinguished from WT littermates (*Figure 2—figure supplement 1*). We identified 968 and 792 transcripts that were significantly downregulated and upregulated, respectively, in *Zfp281*KO versus WT embryos (*Figure 2A*, *Figure 2—source data 1*). Among the significantly downregulated genes were components of the Nodal signaling pathway (e.g., *Nodal, Foxh1, Cripto, Lefty1, Lefty2*), and genes regionally-restricted in the epiblast and its derivatives (e.g., *Fgf5, Otx2, Gsc*) or the VE (e. g., *Cer1, Lhx1, Dkk1, Hex, Hesx1*) (*Figure 2A*, *Figure 2—figure supplement 2*). We also identified a number of genes that were significantly upregulated in mutant embryos, including *Afp, Patched1 (Ptch1), Gsn* (*Figure 2A*). Gene ontology (GO) analysis for the downregulated genes revealed the top enriched biological process to be A-P axis specification (*Figure 2B*). Gene set enrichment analysis (GSEA) revealed that Nodal was the top-ranking signaling pathway enriched in the downregulated genes (*Figure 2C*). Of note, this gene set is also included in the GO term A-P axis specification. We next performed RT-qPCR analysis on E6.5 embryos, and confirmed differential expression of genes that are components of the Nodal signaling pathway, Anterior/AVE markers, and posterior/lineage markers, as well as a few genes that were upregulated in *Zfp281*KO embryos compared to WT embryos (*Figure 2D*), consistent with our transcriptome data (*Figure 2A*). Together, these data demonstrate that Nodal signaling and A-P axis specification, two key events associated with epiblast maturation, were perturbed in the absence of Zfp281.

## Zfp281 controls hallmark molecular events in the exit from naive pluripotency

Changes in the expression of some of the stage-specific pluripotency-associated genes (*Figure 2*) prompted us to investigate this hallmark molecular event involved in the naive-to-primed transition. Under normal development, the pluripotency factors Sox2 and Oct4 are similarly expressed throughout the naive-to-primed transition. On the other hand, Nanog is rapidly shut down after E4.5 (*Chambers et al., 2003*) and expressed again from E6.0 onwards in the epiblast (*Hart et al., 2004*). Nanog is also downregulated in EpiSCs (*Silva et al., 2009*). Consistent with our RNA-seq data, we noted that Sox2 and Nanog protein levels were unaffected in *Zfp281*KO embryos (*Figure 3A,D* and *Figure 3—figure supplement 1*) or in tetraploid chimeras (*Figure 3—figure supplement 2A*). In contrast, Oct4 protein was significantly downregulated in E6.25 *Zfp281*KO embryos (*Figure 3B,D*). Additionally, Otx2, which is activated during the naive-to-primed transition and critical for activation of epiblast gene-related enhancers (*Buecker et al., 2014*), was expressed at significantly reduced levels (*Figure 3C,D*) and mislocalized at the distal tip of mutant embryos, rather than being anteriorly restricted as in WT (*Figure 3—figure supplement 2B*). This downregulation of Otx2 protein expression was already visible at an earlier stage in *Zfp281*KO embryos and in tetraploid chimeras,

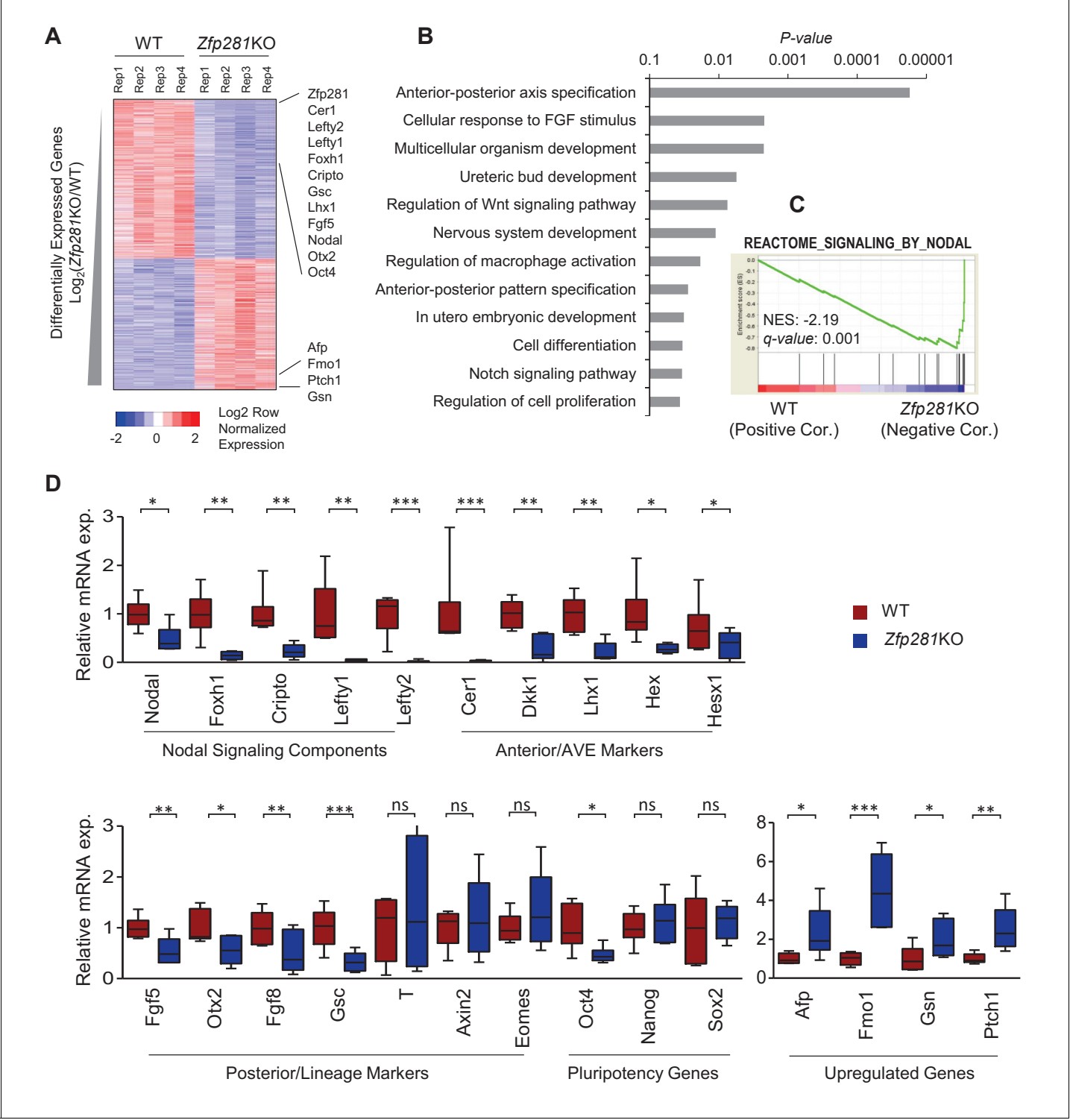

**Figure 2.** *Zfp281*KO embryos exhibit defects in Nodal signaling and expression of genes associated with anterior-posterior patterning. (**A**) Heatmap of genes differentially regulated between WT and *Zfp281*KO embryos (E6.5). (**B**) Gene ontology (GO) analysis for significant downregulated genes (fold-change <4, p-value<0.05) in *Zfp281*KO embryos. (**C**) Gene set enrichment analysis (GSEA) shows Nodal signaling as top pathway in downregulated genes in *Zfp281*KO embryos. (**D**) RT-qPCR for expression of genes at E6.5. For each genotype, n = 6 embryos. mRNA levels of WT embryos were normalized to 1. Student T-test was used examine statistical significance: ns = non significant, *p<0.05, **p<0.01, ***p<0.001. RT-qPCR primer sequences are provided in *Figure 2—source data 2*.

DOI: https://doi.org/10.7554/eLife.33333.006

*Figure 2 continued on next page*

*Figure 2 continued*

The following source data and figure supplements are available for figure 2:

**Source data 1.** RPKM values of significantly up- and down-regulated genes in WT and *Zpf281*KO embryos RNA-seq data.
DOI: https://doi.org/10.7554/eLife.33333.009
**Source data 2.** RT-qPCR primer sequences.
DOI: https://doi.org/10.7554/eLife.33333.010
**Figure supplement 1.** Images of E6.5 WT and *Zpf281*KO embryos used for RNA-seq analysis.
DOI: https://doi.org/10.7554/eLife.33333.007
**Figure supplement 2.** RNA-seq tracks of *Zfp281* and Nodal signaling components.
DOI: https://doi.org/10.7554/eLife.33333.008

together with the downregulation of another marker of primed pluripotency, Oct6 (*Figure 3—figure supplement 2C*), further indicating that epiblast maturation is defective in *Zfp281*KO embryos.

Another hallmark of the naive-to-primed transition is the expression of lineage-specific genes at the posterior part of the embryo, marking the site of the primitive streak and onset of gastrulation. We noted that the T protein is expressed in *Zfp281*KO embryos (*Figure 3E*). However, cells that started expressing T, had not downregulated Nanog (see inset in *Figure 3E*), indicating a failure in the extinguishment of pluripotency-associated genes as epiblast cells committed to differentiate in *Zfp281*KO embryos. The domains of both T and Nanog expression in mutant embryos were also proximally radialized, instead of being posteriorly restricted as in WT embryos (*Figure 3E*). This radialization of normally posteriorly-localized markers was also observed in tetraploid *Zfp281*KO ESCs chimeras (*Figure 3—figure supplement 3A*). Notably, this phenotype was rescued in tetraploid WT <->*Zfp281*KO + Zfp281 cDNA ESC chimeras (*Figure 3—figure supplement 3A*). Analysis of RNA expression by wholemount mRNA in situ hybridization (WISH) of other markers of the primitive streak such as *Fgf8, Axin2* and *Eomes* showed that, similarly to T, they were expressed and proximally radialized in mutant embryos (*Figure 3F* and *Figure 3—figure supplement 3B,C*). Our WISH and RNA-seq data both also revealed that levels of *Fgf8* were reduced upon Zfp281 loss. However, T, Axin2 and Eomes RNA levels were not significantly reduced in *Zfp281*KO embryos (*Figure 2D*). Therefore, Zfp281 is required for induction and proper localization of lineage specification markers.

Together, our data establish a requirement for Zfp281 in controlling key molecular events involved in the naive-to-primed transition in embryos, including extinguishment of pluripotency-associated and induction (or localization) of lineage specification transcriptional programs during epiblast maturation.

## Zfp281 plays a critical role in Nodal signaling activation to promote DVE/AVE migration

Execution of the naive-to-primed transition in the epiblast of early post-implantation embryos culminates in A-P axis formation, determined via cross-talk between the epiblast and VE through Nodal and Wnt signals (*Kiecker et al., 2016*). The radialized expression of Nanog and several lineage specification factors in *Zfp281*KO embryos (*Figure 3E–F*, *Figure 3—figure supplement 2B* and *Figure 3—figure supplement 3A–B*), as well as deregulation of genes involved in A-P axis specification (*Figure 2*) prompted us to examine the expression and localization of AVE/DVE markers. WISH of *Hesx1, Dkk1, Cer1, Hex,* and *Lefty1* revealed their absence or significantly reduced expression in *Zfp281*KO embryos (*Figure 4A* and *Figure 4—figure supplement 1*). WISH for *Dkk1* of *Zfp281*KO tetraploid chimera confirmed this phenotype and showed a partial rescue using *Zfp281*KO + Zfp281 cDNA ESCs (*Figure 4—figure supplement 2A*). In the few cases where expression was detected (e. g., *Hesx1* in *Zfp281*KO or *Dkk1* in tetraploid mutant chimera), it was localized at the distal tip of embryos, but not anteriorly as in WT (*Figure 4A*, *Figure 4—figure supplement 1* and *Figure 4—figure supplement 2A*), suggesting that the DVE/AVE population was not specified, or could not be maintained or migrate. To distinguish these possibilities, we employed a *Hex-GFP* reporter line (*Rodriguez et al., 2001*) to visualize the AVE in *Zfp281*KO embryos. In agreement with our WISH data, *Hex*-GFP expression was reduced and distally-localized, consistent with a failure in maintenance leading to impaired migration of the DVE/AVE population (*Figure 4B* and *Figure 4—figure*

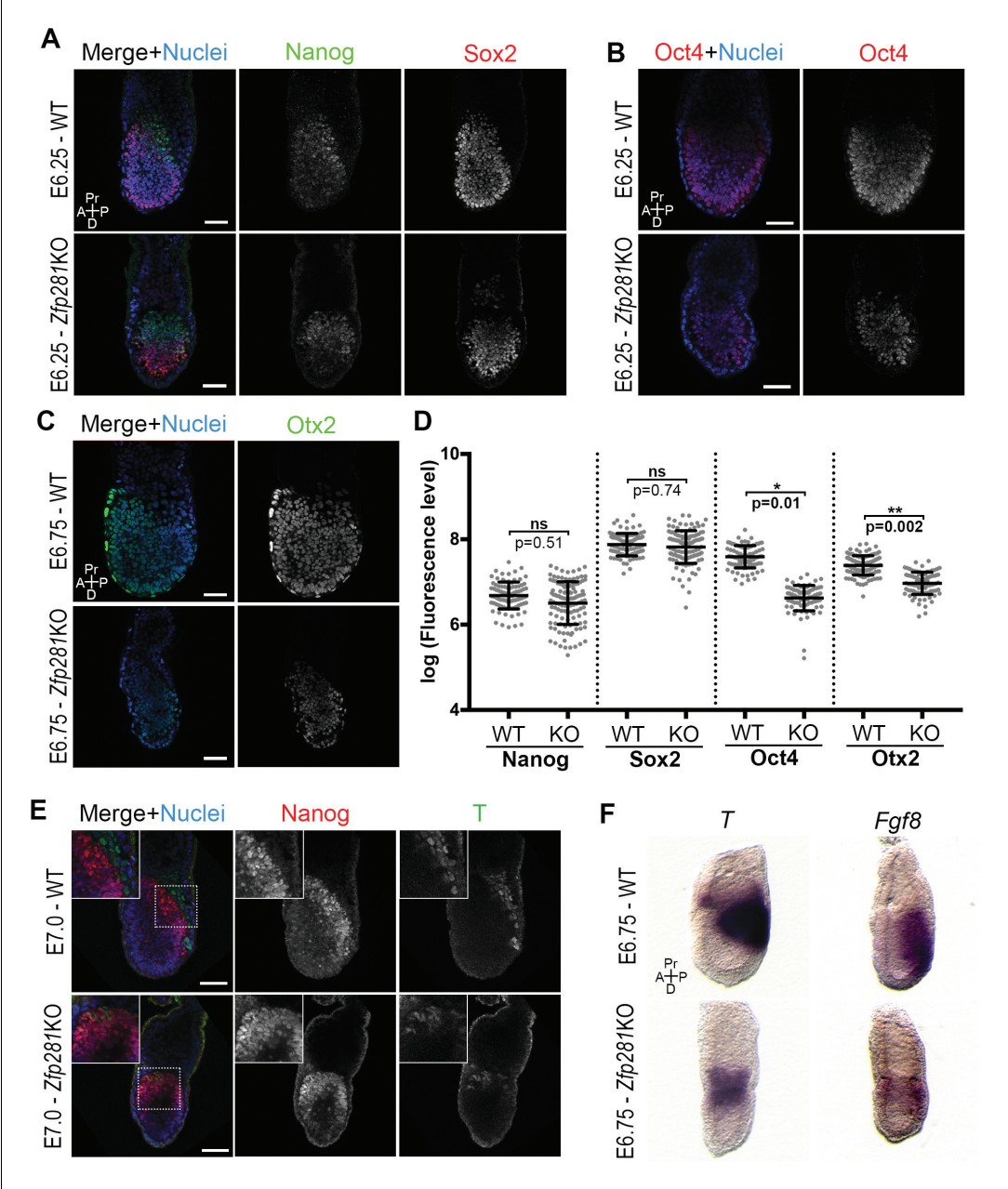

**Figure 3.** Zfp281 regulates pluripotency-associated factors and Otx2. (**A**) Immunostaining of Nanog and Sox2 in WT and *Zfp281*KO embryos (E6.25). (**B**) Immunostaining of Oct4 in WT and *Zfp281*KO embryos (E6.25). (**C**) Immunostaining of Otx2 in WT and *Zfp281*KO embryos (E6.75). (**D**) Fluorescent intensity quantification of Nanog, Sox2, Oct4 and Otx2 proteins (each dot representing the mean corrected fluorescence level per epiblast cell) using Imaris software. For each genotype, n = 3 embryos (30 cells quantified per embryo). Statistical significance was calculated on the average level of corrected fluorescence per embryo using Student T-test. (**E**) Immunohistochemistry of pluripotency factor Nanog which is localized to posterior epiblast in WT embryos, and the primitive streak marker T, exhibit radialized expression in mutant compared to WT (E7.0). Additionally, in *Zfp281*KO, the mutually-exclusive pattern of Nanog and T observed in WT is lost and proteins colocalize in a subset of cells. High magnification insets (top-left) show distribution in regions highlighted. (**F**) WISH of E6.75 *Zfp281*KO and WT littermate embryos. Markers of the primitive streak T and Fgf8 are radialized in *Zfp281*KO embryos compared to WT. A = Anterior, p=Posterior, Pr = proximal, D = Distal, Scale bars represent 50 μm.

DOI: https://doi.org/10.7554/eLife.33333.011

The following figure supplements are available for figure 3:

**Figure supplement 1.** Sox2 and Nanog immunostaining in WT and *Zfp281*KO embryos.

DOI: https://doi.org/10.7554/eLife.33333.012

**Figure supplement 2.** Expression of pluripotency markers in tetraploid chimera and early postimplantation WT and *Zfp281*KO embryos.

*Figure 3 continued on next page*

*Figure 3 continued*

DOI: https://doi.org/10.7554/eLife.33333.013

**Figure supplement 3.** WISH analysis of Otx2, Axin2, and Eomes in WT and *Zfp281*KO embryos.

DOI: https://doi.org/10.7554/eLife.33333.014

*supplement 2B*). By contrast, the expression of proteins labeling the entire VE, such as *Gata6*, *Lhx1* and *Eomes*, was unaffected in *Zfp281* mutants (*Figure 4—figure supplement 2C,D*).

Nodal and Wnt signaling pathways play a key role in DVE/AVE specification and migration (*Kiecker et al., 2016*), and rank as top enriched GO terms in the downregulated genes from our RNA-seq data (*Figure 2C*). We already showed that levels of *Axin2* and *T*, two Wnt signaling targets, were unaffected by loss of Zfp281, although their expression was radialized due to failure in A-P axis specification (*Figures 2D* and *3F* and *Figure 3—figure supplement 3B*). By contrast, the Nodal target and negative regulator *Lefty2* (expressed in the posterior epiblast of WT embryos) were absent in the epiblast of *Zfp281*KO embryos (*Figure 4C*), similarly to the related *Lefty1* gene, which is expressed earlier in the VE, and also absent in mutants (*Figure 4A)*. Nodal and Eomes are usually posteriorly localized in the epiblast at the site of the primitive streak (*Nowotschin et al., 2013*). However, their expression was radialized in *Zfp281*KO embryos (*Figure 4C* and *Figure 3—figure supplement 3B*) while Nodal expression levels were reduced (*Figure 2D* and *Figure 4—figure supplement 2E*).

Together, our data suggest an impairment of Nodal signaling in *Zfp281*KO embryos, leading to defects in DVE/AVE specification and migration, further corroborating a critical role for Zfp281 in promoting epiblast maturation.

## Zfp281 regulates lineage-specific genes for transcriptional activation during epiblast maturation

To understand the molecular mechanisms by which Zfp281 promotes epiblast maturation, we turned to the in vitro naive-to-primed transition model to identify Zfp281-regulated genes. We first examined expression of Zfp281-regulated genes in WT and *Zfp281*KO ESCs and epiblast-like cells (EpiLCs), which is an alternative of primed cells because *Zfp281*KO is detrimental to the self-renewal of EpiSCs (*Tsakiridis et al., 2014*; *Fidalgo et al., 2016*). EpiLCs are derived by short-time (48 hr) adaption of ESCs to the primed cell culture condition, representing an intermediate state, known as the formative state (*Kalkan and Smith, 2014*; *Smith, 2017*), in the naive-to-primed transition (*Hayashi et al., 2011*; *Buecker et al., 2014*). As expected, EpiLCs (WT) expressed higher levels of lineage-specific genes than ESCs (*Figure 5A*). However, levels of expression of these genes were reduced or abrogated in *Zfp281*KO relative to WT EpiLCs (*Figure 5A*), indicating Zfp281 is important for establishing or maintaining expression of these genes during the transition. In addition, while genes involved in Nodal signaling were similarly expressed in both WT ESCs and EpiLCs, their levels of expression were downregulated in *Zfp281*KO versus WT EpiLCs (*Figure 5—figure supplement 1*), consistent with their downregulation in *Zfp281*KO versus WT embryos (*Figure 2A,D*).

To investigate how Zfp281 exerts transcriptional control of downstream target genes, we performed chromatin immunoprecipitation (ChIP) with massively parallel sequencing (ChIP-seq) in both WT ESCs and EpiSCs, which revealed enrichment of Zfp281 binding at regions near gene transcription start sites (TSSs) and enhancers (*Figure 5—figure supplement 2A,B*), suggesting that Zfp281 is actively involved in transcriptional regulation in both naive and primed pluripotent states. There were 9358 common peaks for Zfp281, and 11,408 and 3467 peaks specific to ESCs and EpiSCs, respectively, which were lost and gained during the transition between these two states (*Figure 5—figure supplement 2C,D*), suggesting Zfp281 targets may be dynamically regulated during pluripotent state transition.

Next, we determined whether Zfp281 coordinately controls transcriptional programs associated with pluripotent state transition and lineage commitment through binding of regulatory regions of its target genes. Surprisingly, promoters (Pro) of lineage-specific genes such as *T, Otx2, Eomes, Gsc* were bound by Zfp281 with a higher enrichment in ESCs than in EpiSCs (*Figure 5B*). ChIP-qPCR analyses confirmed that Zfp281 binding intensities were reduced at these promoters in EpiSCs (*Figure 5C*). Consistent with the fact that these lineage-specific genes will be activated in primed

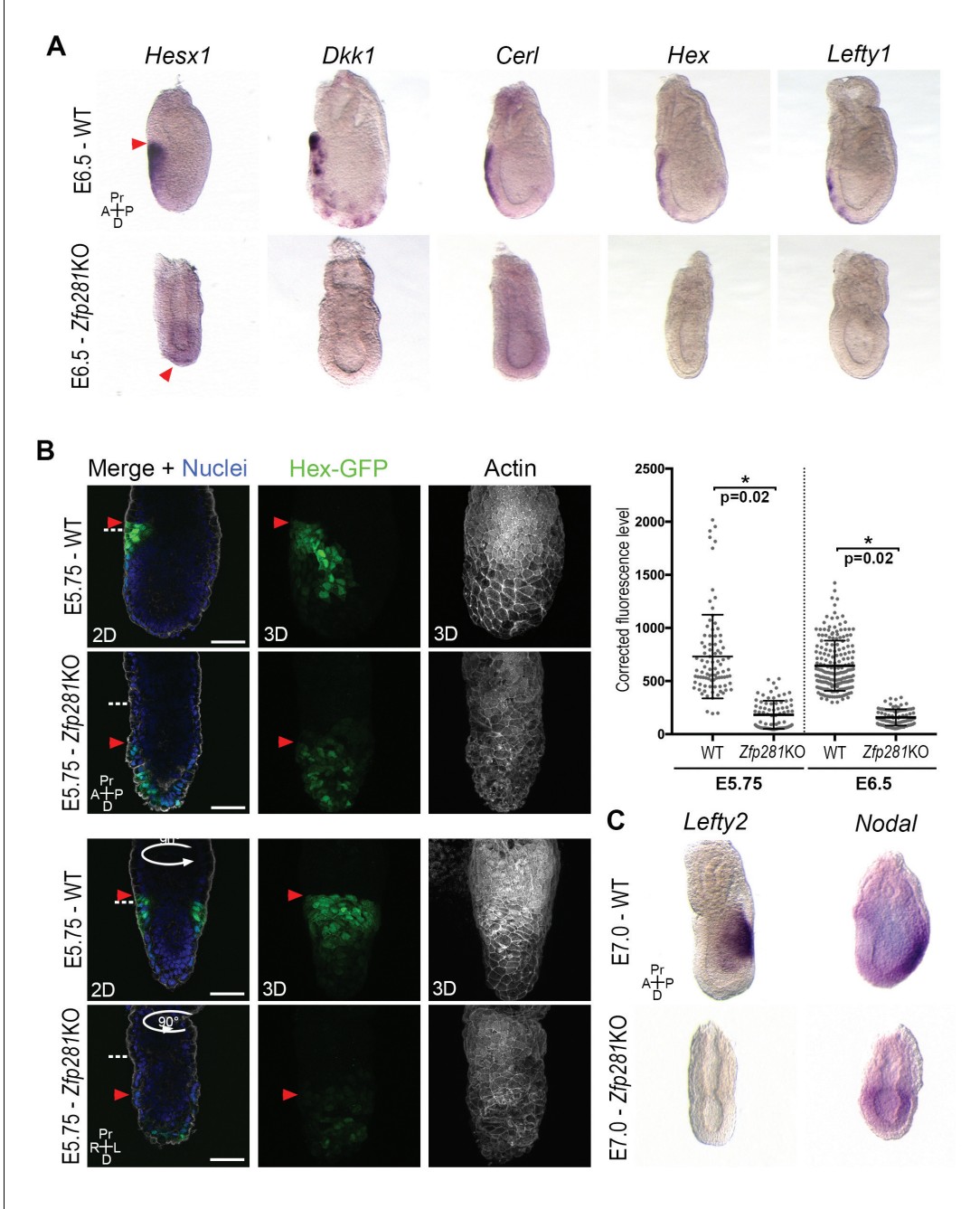

**Figure 4.** Loss of Zfp281 affects pluripotent state progression and results in failure in anterior-posterior axis establishment. (**A**) WISH of E6.5 *Zfp281*KO and WT littermate embryos. Markers of DVE/AVE such as Hesx1, Dkk1, Cer1, Hex and Lefty1 are not expressed in *Zfp281*KO embryos compared to WT littermates, which exhibit distal-anterior domains of expression. Red arrowheads indicate the position of most anterior cells of the DVE/AVE. (**B**) Reduced expression and failure of anterior migration of Hex-GFP reporter in *Zfp281*KO embryos. Cell shape and organization within the VE layer is affected by absence of Zfp281 as indicated by distribution of F-Actin. White dashed line indicates the limit between extra-embryonic and embryonic regions. Quantification of nuclear levels of Hex-GFP at E5.75 (n = 2 embryos, 89 cells for WT and n = 3 embryos, 78 cells for *Zfp281*KO) and at E6.5 (n = 2 embryos, 193 cells for WT and n = 2 embryos, 89 cells for *Zfp281*KO) using Imaris software. Statistical significance was calculated on the average level of corrected fluorescence per embryo using Student's T-test. *p<0.05. (**C**) WISH analysis of the Nodal target Lefty2 shows loss of expression in *Zfp281*KO embryos. Nodal is radialized in *Zfp281*KO embryos compared to WT due to lack of A-P axis. A = Anterior, p=Posterior, Pr = Proximal, D = Distal, 2D = single optical section, 3D = projection of several optical section. Scale bars represent 50 μm.

DOI: https://doi.org/10.7554/eLife.33333.015

The following figure supplements are available for figure 4:

*Figure 4 continued on next page*

*Figure 4 continued*

**Figure supplement 1.** Table of markers and number of embryos analyzed by WISH.
DOI: https://doi.org/10.7554/eLife.33333.016
**Figure supplement 2.** Additional characterization of *Zfp281*KO embryos.
DOI: https://doi.org/10.7554/eLife.33333.017

cells, we also observed diminished intensities of repressive histone mark H3K27me3 on their promoters (*Figure 5D*). By contrast, there was no Zfp281 peak at promoters of *Fgf5* or *Fgf8*, indicating a promoter-independent regulation of Zfp281 on these two genes (*Figure 5E*). However, Zfp281 bound at promoter-distal regions of *Fgf5* and *Fgf8* (*Figure 5E–F*) accompanied with increased H3K27ac (*Figure 5G*) in EpiSCs, suggesting Zfp281 may be involved in enhancer activation on these two targets during the naive-to-primed transition. Indeed, a previous study has shown that the two Zfp281 peaks (P1, P2) comprising enhancers of *Fgf5* are critical for the naive-to-primed transition (*Buecker et al., 2014*). Taken together, our data indicate that Zfp281 regulates lineage-specific genes during epiblast maturation through both promoter- and enhancer-related mechanisms.

## Zfp281 is associated with chromatin modifiers for promoter activation of lineage-specific genes

To further understand how Zfp281 controls transcription of target genes in relation to their promoter chromatin architecture, we investigated the genome-wide association of Zfp281 with other epigenetic regulators and TFs in ESCs. Hierarchical clustering analysis for ChIP-seq association revealed that Zfp281 and Ep400 have the most similar binding patterns. Furthermore, Zfp281 and Ep400 also show similar binding patterns with the Polycomb repressive complex 2 (PRC2) components Ezh2 and Suz12 (*Figure 6A*). Ep400 is a component of the Tip60-Ep400 histone acetyltransferase complex that is necessary to maintain ESC self-renewal (*Fazzio et al., 2008*; *Chen et al., 2015*). A previous study showed that Ep400 localization to promoters depends on H3K4me3, and Ep400 promotes histone H4 acetylation at both active and silent target promoters in ESCs (*Fazzio et al., 2008*). We profiled ChIP-seq intensities of Ep400, Suz12 (a component of PRC2 that modifies H3K27me3), Mbd3 and histone marks H3K4me3, H3K27ac, and H3K27me3 at Zfp281 peak regions (*Figure 6B*). Mbd3 is a core component of NuRD histone deacetylation complex that can be recruited by Zfp281 for repression of pluripotency genes (*Fidalgo et al., 2012*). Zfp281 and Ep400 peaks exhibited a high correlation across the genome, but were mutually exclusive with Suz12 and Mbd3 peaks, dividing Zfp281 peaks into two classes: (I) Zfp281/Ep400/Suz12-cobound, and (II) Zfp281/Ep400/Mbd3-cobound (*Figure 6B*). Class I and Class II characterize the epigenetic features of target genes in ESCs, and are highly enriched for GO terms that signify development and pluripotency, respectively (*Figure 6—figure supplement 1*). The Class I genes are bivalent with a feature of H3K4me3 and H3K27me3 co-enrichment (*Figure 6B*). Bivalent genes remain silent in ESCs while undergoing fast activation in response to differentiation signals (*Bernstein et al., 2006*). Our study demonstrates that Zfp281 regulates active and bivalent genes by associating with identical (Ep400 for both Class I and Class II) or distinct (PRC2 for Class I versus NuRD for Class II) epigenetic regulators.

We next asked whether Zfp281 associates with these epigenetic modifiers during the naive-to-primed transition. While we confirmed interactions of the PRC2 (Suz12) and NuRD (Chd4, Mbd3) complexes with Zfp281 in ESCs, we found their associations with Zfp281 were reduced in EpiSCs despite their expression levels being similar in the two populations (*Figure 6C*). By contrast, Zfp281's interaction with components of the Tip60-Ep400 complex (Ep400, Trrap) was maintained in both ESCs and EpiSCs (*Figure 6C*). Loss of interaction between Zfp281 and the PRC2 complex may be responsible for the activation of Zfp281 target genes during the transition, providing a parsimonious explanation for the upregulation of lineage specification genes (*Figure 5B–C*) and reduction of H3K27me3 (*Figure 5D*) in EpiSCs versus ESCs. The lack of association between Zfp281 and NuRD in EpiSCs may also explain why pluripotency genes including *Sox2* and *Nanog* are not regulated by Zfp281 in the epiblast (*Figure 3A,D*), which is in contrast with NuRD-mediated Nanog repression in ESCs due to their physical association (*Fidalgo et al., 2012*). We confirmed the reduced chromatin occupancy of Suz12 at bivalent promoters of *T* and *Eomes* in *Zfp281*KO relative to WT ESCs (*Figure 6D–E*), indicating a Zfp281-dependent recruitment of PRC2 at the bivalent promoters. However, we cannot exclude the possibility that the residual binding of Zfp281 to bivalent promoters (e.

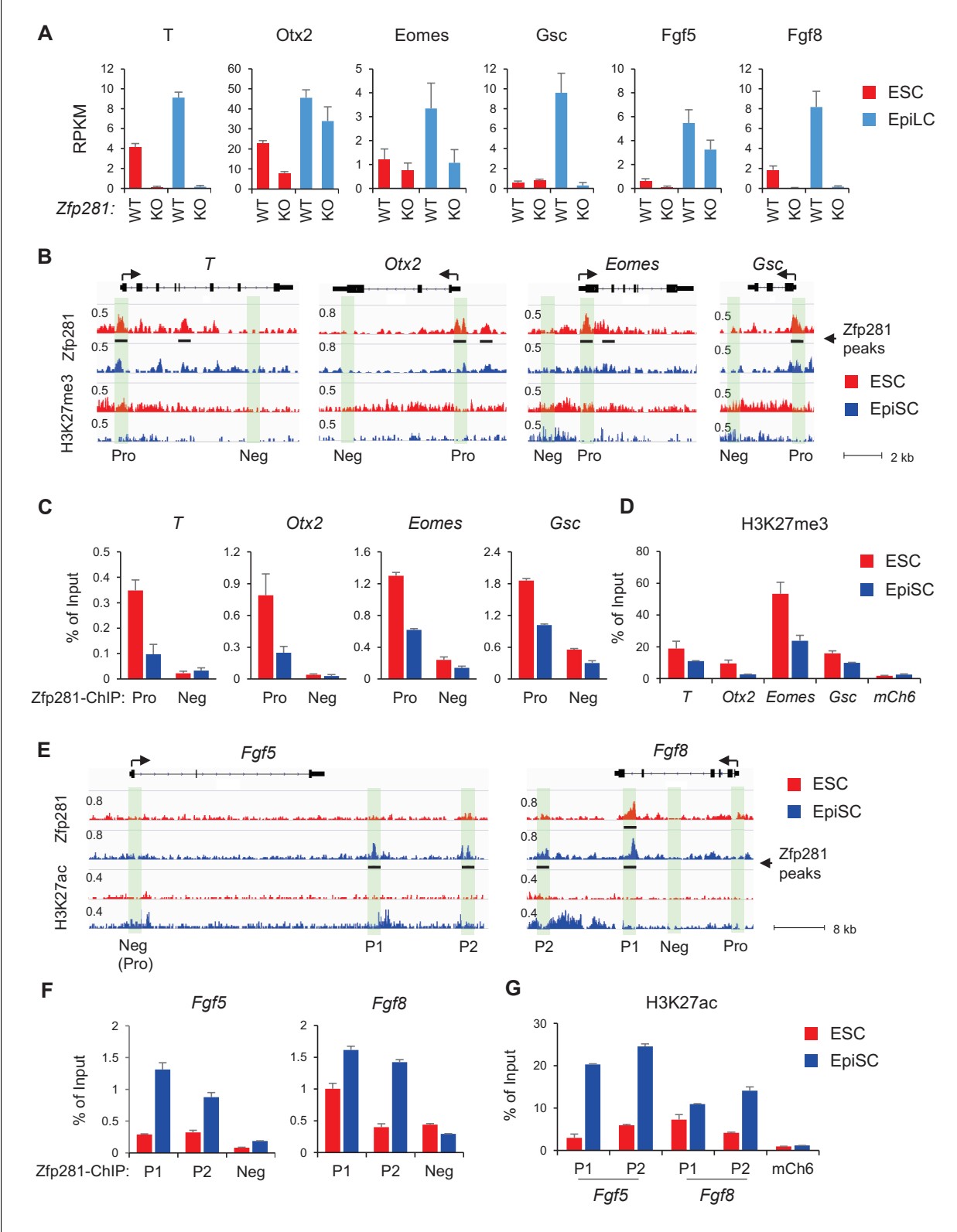

**Figure 5.** Zfp281 activates lineage-specific genes during epiblast maturation. (**A**) Expressions of lineage-specific genes in WT and *Zfp281*KO ESCs and EpiLCs, from a previous published RNA-seq dataset (*Fidalgo et al., 2016*). mRNA expression level is mapped reads per million total mapped reads per kilobase (RPKM). (**B**) Zfp281 and H3K27me3 ChIP-seq tracks at loci of *T, Otx2, Eomes,* and *Gsc.* Black bar under track indicates region called as Zfp281 peak by MACS software. (**C**) ChIP-qPCR of Zfp281 at promoters (Pro) of *T, Otx2, Eomes,* and *Gsc* in ESCs and EpiSCs. Promoter and negative regions

*Figure 5 continued on next page*

Figure 5 continued

for PCR at each gene are shown in panel B. (**D**) ChIP-qPCR of H3K27me3 at promoters of *T, Otx2, Eomes,* and *Gsc. mCh6* is a negative control mapping to a gene desert region on chromosome 6 (*Boyer et al., 2006*). (**E**) Zfp281 and H3K27ac ChIP-seq tracks at loci of *Fgf5* and *Fgf8.* Black bar under track indicates region called as Zfp281 peak by MACS software. (**F**) ChIP-qPCR of Zfp281 at regulatory regions of *Fgf5* and *Fgf8* in ESCs and EpiSCs. Positions for PCR at each gene are shown in panel E. (**G**) ChIP-qPCR of H3K27me3 at regulatory regions of *Fgf5* and *Fgf8. mCh6* is a negative control mapping to a gene desert region on chromosome 6 (*Boyer et al., 2006*). ChIP-qPCR primer sequences are provided in *Figure 5—source data 1*.

DOI: https://doi.org/10.7554/eLife.33333.018

The following source data and figure supplements are available for figure 5:

**Source data 1.** ChIP-qPCR primer sequences.
DOI: https://doi.org/10.7554/eLife.33333.021
**Figure supplement 1.** Expression of Nodal signaling component and pluripotency genes in WT and *Zfp281*KO cells.
DOI: https://doi.org/10.7554/eLife.33333.019
**Figure supplement 2.** Characterization of Zfp281 ChIP-seq in ESCs and EpiSCs.
DOI: https://doi.org/10.7554/eLife.33333.020

g., *T* and *Eomes* in *Figure 5C*) with maintained Tip60-Ep400 complex association during the naive-to-primed transition may further reinforce the activation of lineage-specific genes through the presumed role of Tip60-Ep400 in outcompeting PRC2, and thus downregulating the promoter H3K27me3 levels (*Chen et al., 2015*).

Taken together, our data reveal an important role of Zfp281 in regulating bivalent promoters during the naive-to-primed pluripotency transition. Reduced association between Zfp281 and the PRC2 complex, but preservation of the Zfp281-Ep400 association, at bivalent promoters results in decreased H3K27me3 during this transition, leading to transcriptional activation of lineage-specific genes concomitant with epiblast maturation.

## Zfp281 cooperates with Oct4 and P300 for regulation of Nodal signaling components in epiblast maturation

We have shown that Zfp281 is necessary for activation of Nodal signaling components in the embryo (*Figure 2* and *Figure 4*). Next, we investigated activity of Nodal signaling by examining Smad2 phosphorylation (p-Smad2) in WT and *Zfp281*KO ESCs. P-Smad2 is significantly reduced in *Zfp281*KO ESCs compared to that in WT ESCs (*Figure 7A*). Since *Zfp281*KO EpiSCs cannot be maintained in long-term culture (*Fidalgo et al., 2016*), we performed shRNA-mediated knockdown (KD) of Zfp281 in EpiSCs. P-Smad2 is not affected by *Zfp281*KD in EpiSCs, probably because of the Activin in culture constitutively activating p-Smad2. However, protein expression of the Nodal signaling target Lefty significantly decreased by *Zfp281*KD (*Figure 7—figure supplement 1A*), suggesting that Zfp281 may directly regulate Lefty expression. Similarly, reduction of Lefty protein expression is reproduced by treatment of ALK receptor inhibitor (ALKi) that specifically blocks p-Smad2 and Nodal signaling (*Figure 7—figure supplement 1B*). In addition, as activation of WNT and Nodal signaling pathways can differentiate ESCs to primitive-streak (PS)-like cells (*Mulas et al., 2017*), we also investigated the role of Zfp281 in this differentiation. ESCs were treated with Activin (a Nodal ligand) and CHIR (a GSK3β inhibitor to activate WNT pathway), and with ALKi or Zfp281 shRNAs. The morphology of *Zfp281*KD cells revealed a strong phenotype of differentiation resistance, which is similar to that of ALKi treatment. Dome-shaped ESC-like colonies persisted in *Zfp281*KD cell cultures after Activin/CHIR treatment, a striking difference compared to WT control cells (*Figure 7—figure supplement 2A*). RT-qPCR for up to 3 days after treatment indicated both *Zfp281*KD- and ALKi-treated cells exhibited decreased expression of the PS marker genes *T* and *Lefty2* (a Nodal signaling target gene also expressed in PS) (*Figure 7—figure supplement 2B,C*).

To understand the molecular regulation of the Nodal signaling pathway by Zfp281, we employed ChIP-seq analysis and revealed that Zfp281 localizes to distal regions of *Nodal* and *Lefty2*, as well as promoters of *Foxh1* (*Figure 7C*). Previous studies indicated that transcription factors Oct4, Otx2, and histone acetyltransferase P300 (the writer of H3K27ac) associate with enhancer reorganization in the naive-to-primed transition (*Buecker et al., 2014*; *Yang et al., 2014*). We first confirmed that Zfp281 interacts with Oct4 and P300 in EpiSCs (*Figure 7B*). However, we did not detect the Zfp281-Otx2 interaction in EpiSCs (data not shown). Furthermore, ChIP-seq analysis indicates that Zfp281

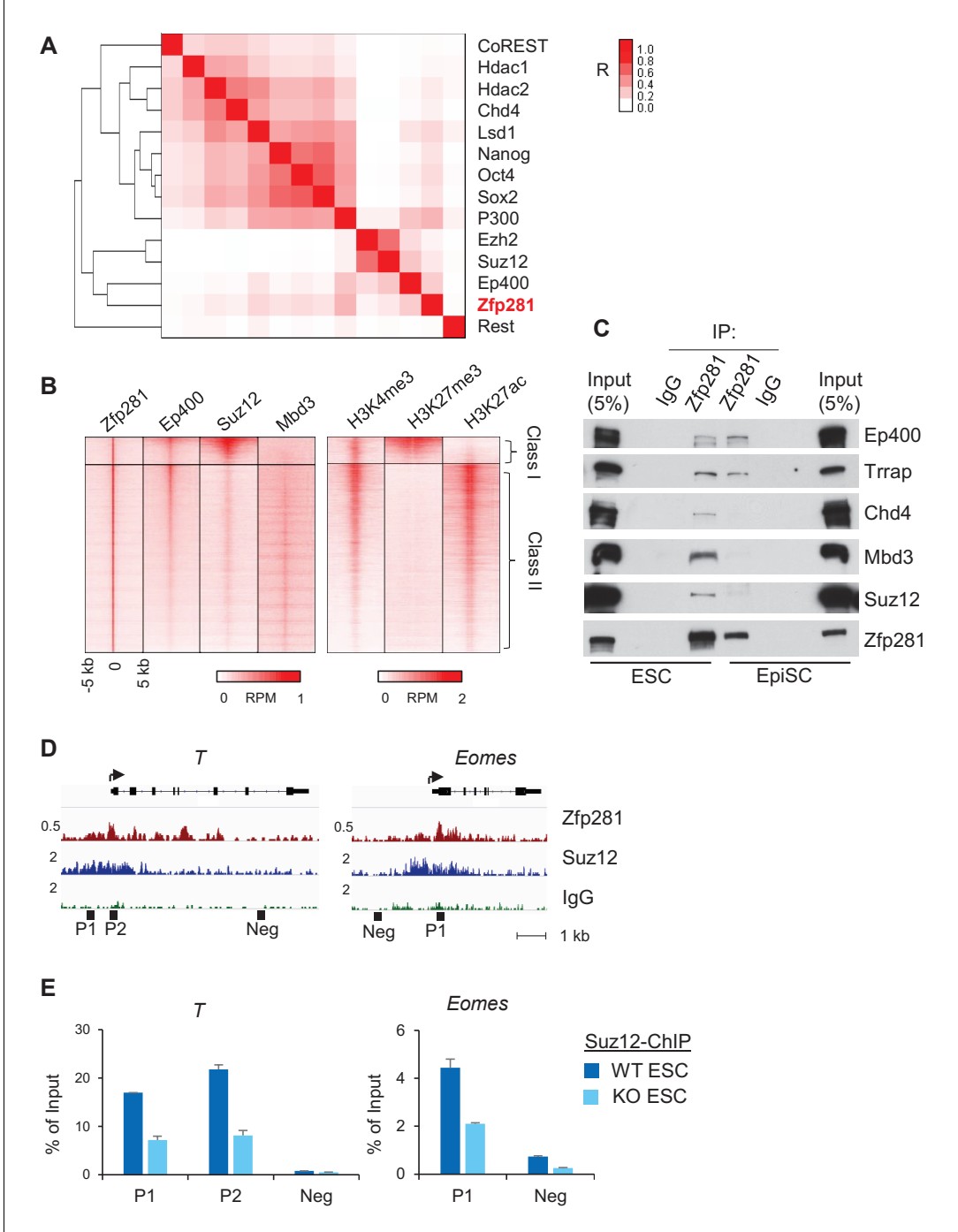

**Figure 6.** Zfp281 associates with PRC2 and Ep400 complexes for promoter regulation of bivalent lineage genes. (**A**) Heatmap of ChIP-seq association analysis of Zfp281 with other transcription factors and epigenetic regulators in ESCs. The ChIP-seq datasets were compiled from published studies (see *Figure 6—source data 1* for the accessions). The scale represents Phi correlation coefficient. (**B**) ChIP-seq profiles of Zfp281, Ep400, Suz12, Mbd3, and histone marks H3K4me3, H3K27me3, and H3K27ac at Zfp281 peak regions. Intensity is plotted within 5 kb around Zfp281 peak center. Two classes of Zfp281 peaks are shown: Class I, Zfp281-Mll2-Ep400-Suz12 co-bound; Class II, Zfp281-Mll2-Ep400-Mbd3 co-bound. (**C**) Co-immunoprecipitation (co-IP) of Zfp281 with other epigenetic regulators in ESCs and EpiSCs. (**D**) ChIP-seq tracks of Zfp281 and PRC2 complex (Suz12) at bivalent *T* and *Eomes* promoters. (**E**) ChIP-qPCR shows Suz12 binding at *T* and *Eomes* promoters decreases in *Zfp281*KO ESCs.
DOI: https://doi.org/10.7554/eLife.33333.022

The following source data and figure supplement are available for figure 6:

**Source data 1.** Accession numbers of ChIP-seq data used in *Figure 6A*.

*Figure 6 continued on next page*

*Figure 6 continued*

DOI: https://doi.org/10.7554/eLife.33333.024

**Figure supplement 1.** GO analysis of Class I and II of Zfp281 targets.

DOI: https://doi.org/10.7554/eLife.33333.023

co-localizes with Oct4 and P300 (*Buecker et al., 2014*) in almost all Zfp281 peaks (*Figure 7C*, *Figure 7—figure supplement 3*), suggesting that Zfp281 may be involved in widespread relocation of Oct4 and P300, including both promoters and enhancers, in the naive-to-primed transition, that is independent of Otx2 association.

Zfp281 localizes to characterized enhancers within the *Nodal* locus (PEE, NDE, HBE, but not ASE, nomenclature from [*Papanayotou et al., 2014*]) (*Figure 7C*). ChIP-qPCR analysis confirmed that Zfp281 intensities are comparable between PEE and NDE, but decreased at HBE in EpiSCs compared to ESCs (*Figure 7D*). Zfp281 intensity by ChIP-qPCR was relatively low at ASE, consistent with our ChIP-seq data (*Figure 7C–D*). It is reported that, during the naive-to-primed transition, *Nodal* enhancer activity relocates from the HBE to the ASE (*Papanayotou et al., 2014*). Indeed, H3K27ac intensity was also reduced at HBE (*Figure 7E*), consistent with reduced Zfp281 binding at this locus during the naive-to-primed transition (*Figure 7D*). We also evaluated Zfp281 intensities by ChIP-qPCR at an enhancer of *Lefty2* and the promoter of *Foxh1*, and found high intensities of Zfp281 and H3K27ac, colocalizing with P300 and Oct4 peaks at these loci in ESCs and EpiSCs (*Figure 7C–E*). Zfp281 binding at regulatory regions of these Nodal signaling-related genes is biologically important, as expression of Lefty2 is downregulated by *Zfp281*KD in EpiSCs (*Figure 7—figure supplement 1*), and many of Nodal signaling genes were perturbed in *Zfp281*KO EpiLCs relative to their WT counterparts (*Figure 5—figure supplement 1*), supporting a critical role of Zfp281 in transcriptional activation of Nodal signaling during the transition to primed pluripotency in vitro. To provide direct evidence that Zfp281 binds and regulates these Nodal signaling related genes in vivo, we performed ChIP experiments on E6.5 WT embryos (*Figure 7F*). Zfp281 exhibited high chromatin-binding activity at the HBE enhancer of the *Nodal* locus, as well as the regulatory regions within the *Lefty2* and *Foxh1* loci (*Figure 7G*), which would be abrogated in *Zfp281* mutant embryos. Together, our data demonstrate that Zfp281, together with P300 and Oct4, are important for regulation of components of the Nodal signaling pathway during the maturation of the epiblast of the mouse embryo.

## Discussion

Pluripotency is a continuum where the naive and primed states represent the initial and final stages, corresponding to the establishment of pluripotency in the pre-implantation blastocyst and the exit from pluripotency as cells of the post-implantation epiblast initiate gastrulation. While the in vitro pluripotent state transition model provides a useful tool to infer how the epiblast transitions between states and prepares for germ layer differentiation (*Weinberger et al., 2016*), it cannot substitute for direct investigations into the transition occurring during epiblast maturation in vivo in embryos. Notably, none of the pluripotency-associated factors, for which mouse mutants are available, have described phenotypes directly associated with the exit from naive pluripotency in the embryo. Zfp281 is the first factor demonstrated to be critical in vivo for this developmental transition; its loss results in embryonic lethality due to a failure in epiblast maturation. Our data reveal that Zfp281 is concomitantly required for the initiation of expression of genes encoding Nodal signaling components and the lineage specification program during epiblast maturation. Zfp281 coordinates crosstalk among multiple epigenetic pathways through physical association with histone methylation (PRC2) and acetylation (Ep400, P300) complexes within epiblast cells. Crosstalk converges on the coordinated regulation of bivalent promoters and reorganization of enhancers leading to activation of lineage-specific genes and downregulation of pluripotency genes during the naive-to-primed transition, arguing for a master regulator status of Zfp281 in controlling key molecular events leading to epiblast maturation (*Figure 8*, top panel). Accordingly, loss of Zfp281 results in a series of developmental defects within the epiblast, which non-cell-autonomously lead to a failure to establish or maintain the DVE/AVE resulting in A-P axis specification defects (*Figure 8*, bottom panel).

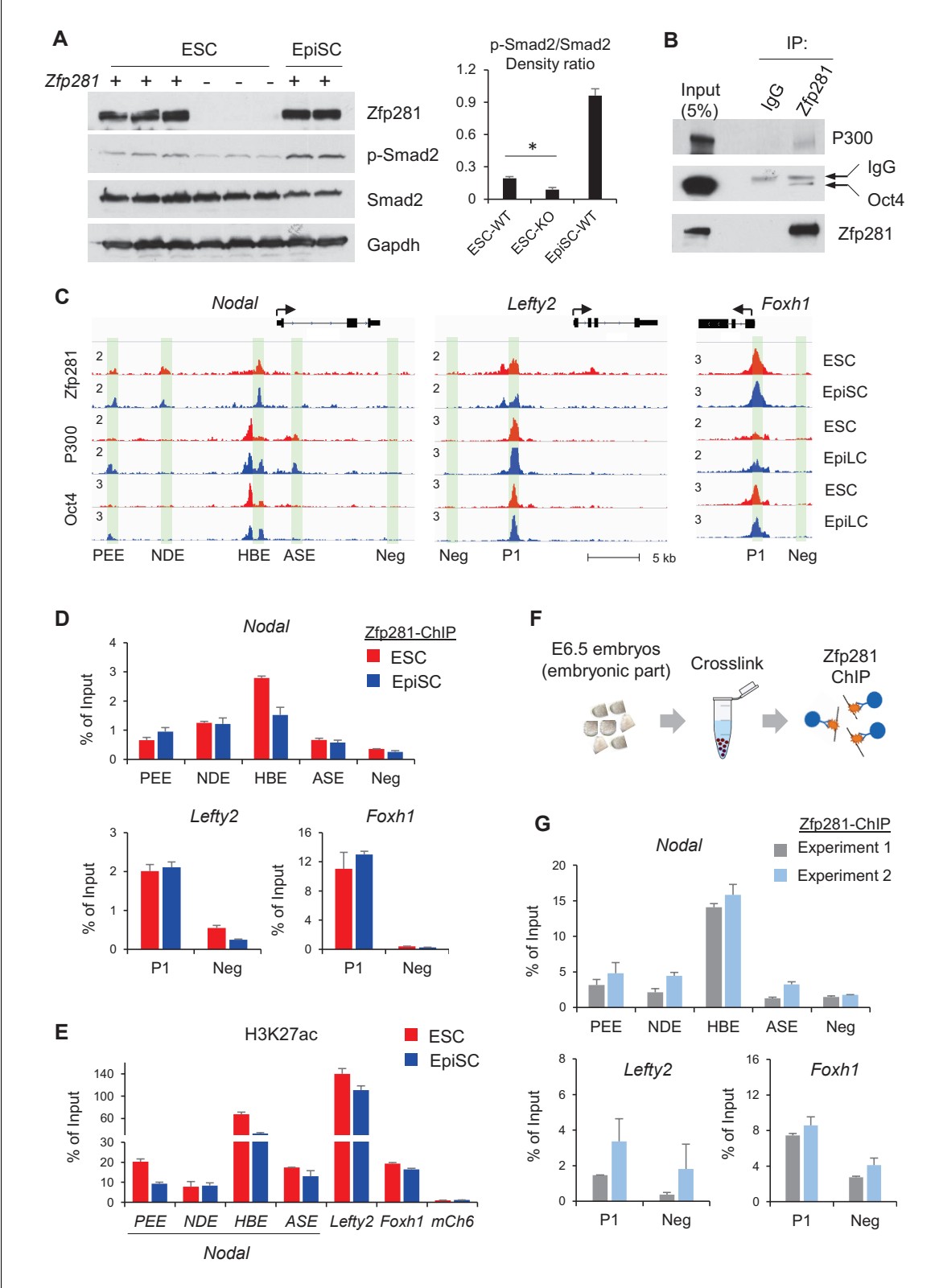

**Figure 7.** Zfp281 cooperates with Oct4 and P300 for regulation of genes encoding Nodal pathway components. (**A**) Protein expression of p-Smad2 and Smad2 in WT (J1, CJ7, WT clone no 3), *Zfp281*KO (clone no. 2.6, 7, 3.34) ESC and WT EpiSC (OEC2, EpiSC9) cells. P-Smad2/Smad2 density ratios were quantified on the right panel. Student T-test was used examine statistical significance, *p<0.05. (**B**) Co-IP of Zfp281 with Oct4 and P300 in EpiSCs. The non-specific band is IgG heavy chain. (**C**) ChIP-seq tracks of Zfp281, Oct4 and P300 in ESCs and EpiSCs/EpiLCs at *Nodal*, *Lefty2*, and *Foxh1* loci.

*Figure 7 continued on next page*

*Figure 7 continued*

Intensities are shown as mapped reads per million total mapped (RPM). (D) ChIP-qPCR of Zfp281 at *Nodal, Lefty2, and Foxh1* regulatory regions in in ESCs and EpiSCs. (E) ChIP-qPCR of histone markers H3K27me at *Nodal, Lefty2, and Foxh1* regulatory regions in ESCs and EpiSCs. (F) Diagram of ChIP experiments on E6.5 (WT) embryos. (G) ChIP-qPCR of Zfp281 at *Nodal, Lefty2, and Foxh1* regulatory regions in E6.5 (WT) embryos. ChIP experiments were performed in two independent replicates. ChIP-qPCR primer sequences are provided in *Figure 7—source data 1*.
DOI: https://doi.org/10.7554/eLife.33333.025

The following source data and figure supplements are available for figure 7:

**Source data 1.** ChIP-qPCR primer sequences.
DOI: https://doi.org/10.7554/eLife.33333.029
**Figure supplement 1.** Zfp281 is required to maintain Lefty expression in vitro.
DOI: https://doi.org/10.7554/eLife.33333.026
**Figure supplement 2.** Zfp281 is required for ESCs differentiating to primitive streak (PS)-like cells.
DOI: https://doi.org/10.7554/eLife.33333.027
**Figure supplement 3.** Zfp281, Oct4, and P300 colocalize in ESCs and EpiSCs/EpiLCs.
DOI: https://doi.org/10.7554/eLife.33333.028

The in vitro pluripotent stem cell models overcome the limitation of embryo material and so have been instrumental in dissecting molecular events involved in the naive-to-primed transition (*Buecker et al., 2014*; *Factor et al., 2014*) as well as the regulation of the Nodal signaling pathway during epiblast maturation (*Papanayotou et al., 2014*; *Papanayotou and Collignon, 2014*). Our combined in vitro and in vivo studies have demonstrated high consistency of the regulatory functions of Zfp281 on the lineage-specific genes (*Figure 2* and *Figure 5*). However, there are also some differences between our findings in embryos and those made in pluripotent stem cell models. For instance, embryo RNA-seq data indicated non-significant changes in expression of lineage markers T and Eomes between WT and *Zfp281*KO embryos (*Figure 2D*), while their expression was markedly downregulated in *Zfp281*KO EpiLCs (*Figure 5A*). This disparity could be due to EpiLC representing a pluripotent state that does not match with the embryonic stage examined for RNA-seq in embryos, and/or extra-embryonic expression, in the case of Eomes in vivo (*Nowotschin et al., 2013*) (*Figure 3—figure supplement 3B*), that is not represented in EpiLC culture. Similarly, *Zfp281*KO embryonic phenotypes described for the VE layer cannot be reproduced in vitro since there is no equivalent cell population in ESC/EpiSC cultures. We noted that the morphology of VE cells in *Zfp281*KO embryos was different from the WT and resembled their adjacent extra-embryonic VE cells (*Figure 1—figure supplement 3*). As we identified that *Afp*, a pan-VE marker whose downregulation is coincident with the onset of gastrulation (*Viotti et al., 2014*), is upregulated in our mutant embryos (*Figure 2A,D*), we speculate that *Zfp281*KO embryos may also fail in VE maturation.

Furthermore, we also observed that Nodal signaling was significantly reduced in *Zfp281*KO embryos (*Figure 2D*), whereas in EpiSCs, p-Smad2 level was not affected by *Zfp281*KD (*Figure 7—figure supplement 1*). We speculate that this discrepancy may be attributable to the presence of Activin in the primed cell culture medium, which constitutively activates Nodal signaling. Compared to the in vitro system, crosstalk between the in vivo epiblast and its adjacent extra-embryonic tissues is dynamic and context-dependent, and unable to be precisely captured in a single defined cell culture system. Moreover, mRNA and protein levels of Oct4 and Otx2, the two factors required for reorganization of the enhancer landscape during the naive-to-primed transition (*Buecker et al., 2014*), are significantly downregulated in *Zfp281*KO embryos (*Figure 2D* and *Figure 3B–D*), which may also explain why Nodal signaling fail to be activated in epiblast maturation (*Figure 7*). Taken together, our studies highlight the importance of direct in vivo functional investigations using mouse models to refine and/or authenticate in vitro findings of pluripotent state transitions for a better understanding of epiblast maturation.

Nodal signaling has been extensively studied in key sequential events ranging from epiblast maturation to left-right patterning (*Shen, 2007*). In *Nodal* mutant embryos, epiblast cells do not mature properly and embryos exhibit a failure in DVE/AVE establishment (*Brennan et al., 2001*; *Mesnard et al., 2006*). A recent study suggests that Nodal guides the transition from naive to formative pluripotency in vitro (Mulas et al., 2017). Although pluripotency factors have been reported to occupy and regulate enhancers at the *Nodal* locus during the ESC-to-EpiSC transition

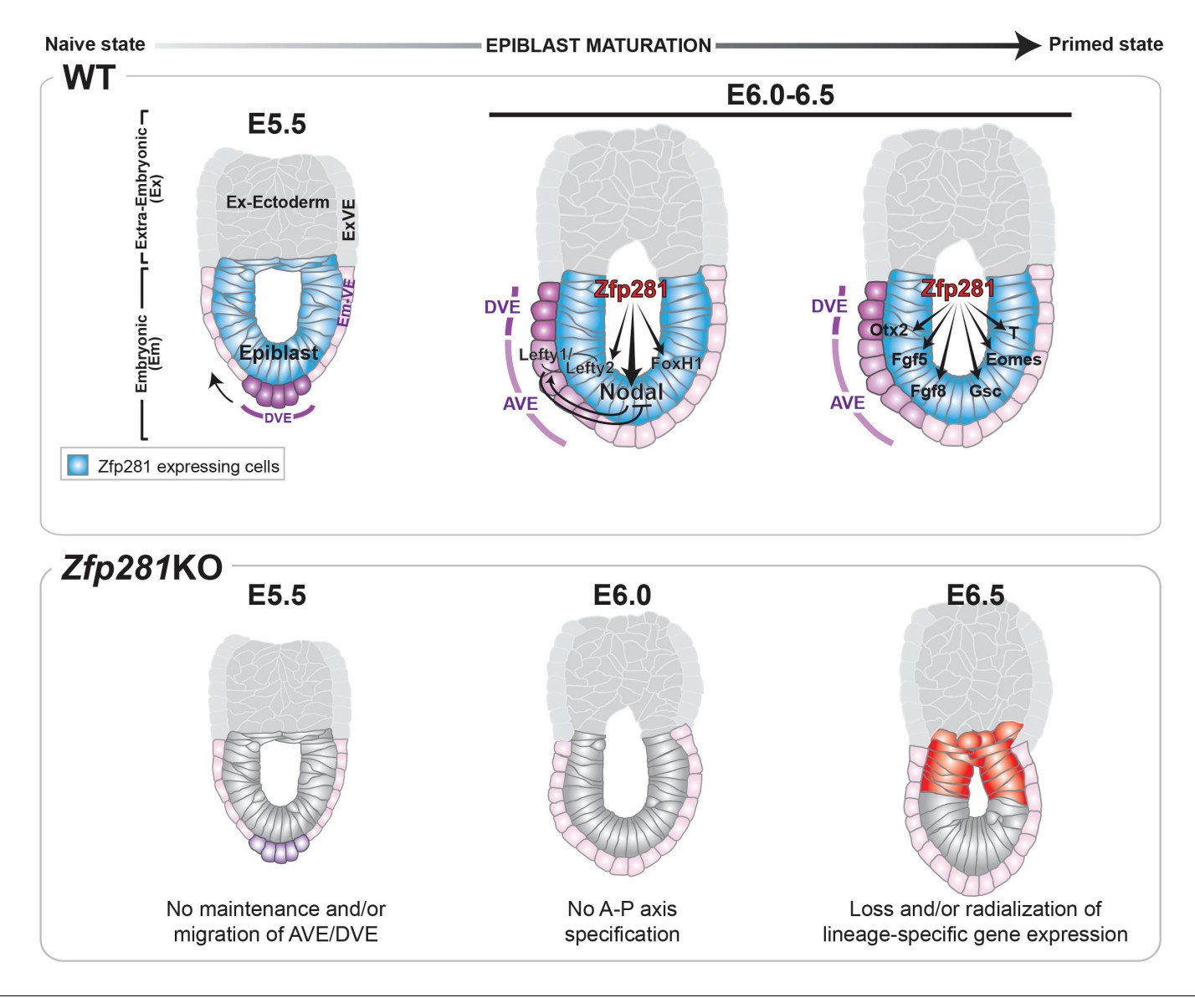

**Figure 8.** Working model for the role of Zfp281 in epiblast maturation. Upper panel describes a working model for Zfp281 function during epiblast maturation in the post-implantation embryo. Zfp281 is expressed in epiblast cells (blue) and directs the activation of target genes. The lower panel describes the *Zfp281*KO embryo phenotype. Defects in epiblast maturation and Nodal signaling lead to failure in anterior-posterior axis establishment. Markers of the posterior epiblast (red) are radialized.

DOI: https://doi.org/10.7554/eLife.33333.030

(*Papanayotou et al., 2014*), how they contribute to lineage specification during the exit from a naive pluripotent state in vivo has remained an open question. Our work reveals that Zfp281 acts as a master regulator of epiblast maturation through the regulation of primed pluripotency genes and Nodal signaling components. Zfp281 binds the PEE, NDE, and HBE *Nodal* enhancers in vitro and in vivo, with binding at the HBE decreasing during epiblast maturation. In ESCs, HBE is bound by Oct4/Sox2/Nanog/Klf4 (*Papanayotou et al., 2014*), as well as Zfp281, however neither Zfp281 nor Oct4 binds ASE (*Figure 7C*), suggesting that loss of Zfp281 binding at HBE may be a prerequisite for decommissioning the naive-specific *Nodal* enhancer in transitioning to the primed state. In addition, Nodal expression is reduced or not restricted proximally in *Zfp281*KO embryos, which could further indicate a failure in enhancer switching. Our findings are thus in agreement with the proposed switch

of *Nodal* enhancers, from HBE to ASE, concomitant with the transition from ESCs to EpiSCs (*Papanayotou et al., 2014*). Together with the crosstalk between Zfp281 and mulitple histone modifying complexes (PRC2/Ep400/NuRD) (*Figure 6*), our data have thus uncovered a critical role for the pluripotency factor Zfp281 in coordinating transcriptional and epigenetic control of the exit from naive pluripotency through promoter and enhancer remodeling, leading to activation of lineage-specific genes and genes encoding Nodal signaling components while embryos are progressing to a primed pluripotency state during epiblast maturation (*Figure 8*).

As the targets and antagonists of Nodal signaling in embryo development, *Lefty* genes (*Lefty1* and *Lefty2*) are sensitive to status of DNA methylation at promoters. It is reported that ESCs depleted of Tet1 diminished expression of Lefty1 with DNA hyper-methylation at *Lefty1* promoter (*Koh et al., 2011*). Our previous work indicated that Zfp281 physically interacts with Tet1 (*Fidalgo et al., 2016*), therefore Zfp281 may recruit Tet1 at *Lefty1/2* promotors and maintain Lefty expression, suggesting a Nodal signaling-independent role of Zfp281. Although, developmental arrest of *Zfp281* mutant embryos (~E6.5) precedes that of Tet-TKO (triple KO) mutants (~E7.5), and defects observed in Tet-TKO mutants were attributed to downregulation of Lefty1/2 through promoter hyper-methylation, leading to constitutive activation of the Nodal signaling pathway due to the disruption of Lefty-mediated negative-feedback (*Dai et al., 2016*). Therefore, these results raise the question of a possible link between Zfp281 and DNA modification for epigenetic control of proper embryonic development, which is worthy of future investigation.

## Materials and methods

### Mouse strains, embryo collection and staining

Mouse strains used in the study were *Zfp281*KO (*Fidalgo et al., 2011*) and *Hex-GFP* (*Rodriguez et al., 2001*). All mice used in this study were maintained in accordance with the guidelines of the Memorial Sloan Kettering Cancer Center (MSKCC) Institutional Animal Care and Use Committee (IACUC) under protocol number 03-12-017 (PI Hadjantonakis).

Pre-implantation embryos were flushed in M2 medium and processed according to standard protocols (*Saiz et al., 2016*). Post-implantation embryos were dissected in DMEM/F12 medium supplemented with 5% newborn calf serum and fixed for 20 min in PBS-4% PFA at room temperature for immunohistochemistry or overnight (o/n) at 4°C for wholemount RNA in situ. Following washes in PBS, embryos were permeabilized in PBS-0.5% Triton for 20 min at room temperature, washed in PBS-0.1% Triton (PBT) and blocked at 4°C o/n in PBT-3% BSA. Primary and secondary antibody stainings were performed o/n at 4°C. Counterstaining with Hoechst and fluorophore-coupled phalloidin (Life Technologies, Carlsbad, CA) were performed for 1 hr at room temperature prior to imaging on a Zeiss LSM880 laser scanning confocal microscope. Details of antibodies used in this study can be found below. Fluorescence intensity levels were measured on data acquired with the same imaging parameters. Pre-implantation fluorescence was quantified with our semi-automated MINS 3D nuclear segmentation software (*Lou et al., 2014*; *Saiz et al., 2016*). Post-implantation nuclear protein levels were quantified using Imaris software (Bitplane) by manually creating individual nuclear surfaces for each cell and quantifying the fluorescence level inside the volume defined by these surfaces. Fluorescence levels were corrected for fluorescence decay along the z axis. Statistical significance was calculated on the average level of corrected fluorescence per embryo using unpaired two-tailed Student T-test (with Welch's correction when standard deviations differed between samples). Each graph represents either corrected fluorescence values per cell or transformed values using natural log (to represent different protein levels on a similar scale). Primary antibodies used for embryo staining were: Zfp281 (Santa Cruz, sc-166933, 1:200, RRID:AB_10612046), Nanog (Reprocell, RCAB002P-F, 1:500, RRID:AB_2616320), Gata6 (R and D, AF1700, 1:100, RRID:AB_2108901), Brachyury/T (R and D, AF2085, 1:100, RRID:AB_2200235), Oct4 (Santa Cruz, sc-9081X, 1:1000, RRID:AB_2167703), Sox2 (R and D, AF2018, 1:50, RRID:AB_355110), Lhx1 (R and D, MAB2725, 1:250, RRID:AB_2135636), Eomes (Abcam, ab183991, 1:500), Otx2 (R and D, AF1979, 1:1000, RRID:AB_2157172), Oct6 (sc-376143, 1:100, RRID:AB_10989975).

## Whole-mount in situ hybridization (WISH)

*Wholemount mRNA in situ hybridizations* were performed according to standard protocols (*Behringer et al., 2014*). The list and citation for each antisense probe used can be found in the following references: *Hesx1* (*Thomas and Beddington, 1996*), *Dkk1* (*Glinka et al., 1998*), *Cerl* (*Belo et al., 1997*), *Hex* (*Thomas et al., 1998*), *Lefty1* (*Oulad-Abdelghani et al., 1998*), *Lefty2* (*Meno et al., 1997*), *Nodal* (*Zhou et al., 1993*), *Otx2* (*Ang and Rossant, 1994*), *T* (*Wilkinson et al., 1990*), *Eomes* (*Ciruna and Rossant, 1999*), *Axin2* (*Jho et al., 2002*), and *Fgf8* (*Crossley and Martin, 1995*).

## ESCs and EpiSCs cell culture

Mouse embryonic stem cells (mESCs) J1 (strain 129S4/SvJae, RRID:CVCL_6412,) and CJ7 (strain 129S1/SvlmJ, RRID:CVCL_C316) were cultured on 0.1% gelatin-coated plates in ESM medium: DMEM supplemented with 15% fetal bovine serum (FBS), 1000 units/mL recombinant leukemia inhibitory factor (LIF), 0.1 mM 2-mercaptoethanol, 2 mM L-glutamine, 0.1 mM MEM non-essential amino acids (NEAA), 1% nucleoside mix (100X stock, Sigma), and 50 U/mL Penicillin/Streptomycin. Primed mouse epiblast stem cells (EpiSCs) were culture on fibronectin-coated plates (10 $\mu$g/mL/cm$^2$) in N2B27 medium supplemented with Activin A (20 ng/mL) and Fgf2 (12 ng/mL) (*Fidalgo et al., 2016*). Generation of *Zfp281*KO ESCs (from CJ7 background, clone no. 2.6 (XX), 3.34 (XY) and 7 (XX)) has been previously described (*Fidalgo et al., 2011*). All cell lines are from authenticated sources and mycoplasma contamination test was performed routinely.

## Tetraploid WT <-> ESCs chimera

Tetraploid chimeras were generated according to standard protocols (*Eakin and Hadjantonakis, 2006*; *Eggan et al., 2001*). Briefly, females were superovulated by i.p. injection of pregnant mares' serum and human chorionic gonadotropin (HCG), and were then mated with males. Fertilized zygotes were isolated from oviducts 24 hr later, cultured until they reached the 2-cell stage, at which point they were electrofused. Fused 1-cell embryos were carefully identified, cultured for another 2 days, and then injected with about fifteen ESCs. ESCs from two *Zfp281*KO clones 2.6 (XX) and 3.34 (XY), and a Zfp281 cDNA rescued 3.34 clone (XY) were used for tetraploid injection. About 40 ~ 60 injected blastocysts were collected for each ES clone, and transferred into 2 ~ 3 pseudo-pregnant foster females. All experiments involving 2C-embryo electrofusion and tetraploid injection were performed at the Rodent Genetic Engineering Laboratory at New York University, with dissections of resulting post-implantation embryo chimeras taking place at MSKCC.

## Knockdown of Zfp281, ALK inhibitor (ALKi) treatment and western blot analysis

Two shRNAs for Zfp281 knockdown were previously validated in our study (*Fidalgo et al., 2016*). Lentivirus production and infection were performed as described (*Ivanova et al., 2006*). Concentrated viral supernatants were incubated with ESCs or EpiSCs for 1 hr, then cells were diluted with fresh medium. Blasticidin (10 $\mu$g/mL) was used for selection 24 hr later, and cells were harvested 72 hr after viral infection.

For ALKi treatment, ESCs and EpiSCs were treated with ALK inhibitor A83-01 (1 $\mu$M, #2939, Tocris Bioscience) for 48 hr with DMSO as a vehicle control. Western blot analysis was carried out using the following primary antibodies: Zfp281 (sc-166933, Santa Cruz, RRID:AB_10612046), p-Smad2 (#3108, Cell Signaling, RRID:AB_490941), Smad2 (#3103, Cell Signaling, RRID:AB_490816), Lefty (sc-365845, Santa Cruz, RRID:AB_10847353, detecting both Lefty1 and Lefty2), Gapdh (10494–1-AP, ProteinTech, RRID:AB_2263076). Blot intensities were quantified using ImageJ software and Student T-test was used to examine statistical significance.

## Induction of primitive streak (PS)-like cells

ESCs were seeded on fibronectin-coated plates, and treated with Activin A (20 ng/mL) and CHIR99021 (3 $\mu$M) for up to 3 days for PS-like cells inductions. Cells were also treated with ALK inhibitor (1 $\mu$M) or Zfp281 shRNAs to investigate the effects of Nodal signaling inhibition or Zfp281 knockdown during this differentiation. RNA was collected at different days after treatment.

## Co-immunoprecipitation (coIP)

Two 15 cm dishes containing comparable numbers of confluent ESCs and EpiSCs were harvested, and nuclear extracts were prepared as described previously (*Costa et al., 2013*). For immunoprecipitation, nuclear extracts of ESCs and EpiSCs were prepared and incubated with pre-bound 4 μg Zfp281 (Abcam, ab101318) or IgG (Millipore, PP64) antibodies with protein G-Agarose beads (#11243233001, Roche) overnight at 4°C. Immunoprecipitates were washed five times with IP buffer, eluted from the beads by boiling, and separated by SDS-PAGE. Western blot analyses were carried out using the following primary antibodies: Zfp281 (sc-166933, Santa Cruz, RRID:AB_10612046), Ep400 (Bethyl, A300-541A, RRID:AB_2098208), Trrap (Santa Cruz, sc-5405, RRID:AB_2209666), Chd4 (Abcam, ab70469, RRID:AB_2229454), Mbd3 (Abcam, ab157464), Suz12 (Abcam, ab12073), Oct4 (Santa Cruz, sc-5279, RRID:AB_628051), and P300 (Santa Cruz, sc-584, RRID:AB_2293429).

## RT-qPCR

Total RNA was extracted from ESCs or EpiSCs using the RNeasy kit (Qiagen), and from E6.5 embryos using the TRIZOL reagent (Ambion, #15596018). Reverse transcription was performed and cDNA was generated using the qScript kit (Quanta, Cat# 95048). Relative expression levels were determined using a LightCycler 480 SYBR green mix (Roche, 4729749001). qRT-PCR experiments were performed on a LightCycler Real Time PCR System (Roche). Gene expression levels were normalized to Gapdh. Error bars indicate standard error for average expression of two technical replicates. Primers for qPCR are listed in *Figure 2—source data 2*.

## ChIP-qPCR

ChIP assays were performed as described (*Lee et al., 2006*). Briefly, cells were cross-linked with 1% (w/v) formaldehyde for 10 min at room temperature, and formaldehyde was inactivated by the addition of 125 mM glycine. For embryo ChIP, the protocol was modified. Briefly, embryonic regions were dissected from 24 (Experiment#1) or 25 (Experiment#2) E6.5 WT embryos. After formaldehyde crosslinking and quenched by glycine, embryos were washed once with lysis buffer #1, then resuspend in lysis buffer #3 for sonication. Sonication was performed on a Bioruptor system, with 30 s ON, 30 s OFF, 30 cycles, high amplitude. Chromatin extracts containing DNA fragments were immunoprecipitated by incubating with primary antibody-conjugated DynaBeads (Novex, 10003D) overnight with rotation at 4°C. The immunoprecipitated DNA was purified with ChIP DNA Clean and Concentrator columns (Zymo Research), analyzed by qPCR using Roche SYBR Green reagents and a LightCycler480 machine, and percentage of input recovery was calculated. Error bars indicate standard error for average expression of two technical replicates. The primary antibodies used for ChIP: Zfp281 (Abcam, ab101318, RRID:AB_11157929), H3K27ac (Abcam, ab4729, RRID:AB_2118291), H3K27me3 (Millipore, 07–449, lot 2194165, RRID:AB_310624), Suz12 (Active Motif, #39357, RRID: AB_2614929), and IgG (Millipore, PP64, RRID:AB_97852). The primers used for qPCR are listed in *Figure 5—source data 1* and *Figure 7—source data 1*.

## RNA-seq analysis of embryos

Mouse embryos were dissected at E6.5 and total RNAs were extracted using TRIZOL reagent (Ambion, #15596018) following standard protocols. RNA quality was evaluated with an Agilent 2100 BioAnalyzer system, and embryo genotype was determined by morphology and confirmed by expression of *Zfp281* by RT-qPCR. Ten to one-hundred ng total RNA from each embryo was processed for RNA-seq library construction using the Ovation Mouse RNA-seq kit (NuGEN, #0348–32) following the manufacturer's protocol, then massively parallel sequencing was performed on an Illumina HiSeq 4000 Sequencing System. Between 20 and 50 million 100 bp single-end reads were obtained per sample.

RNA-seq reads were aligned to the genome using TopHat (v2.0.10) and Bowtie2 (v2.1.0) with the default parameter settings. UCSC mm9 mouse genome, as well as the transcript annotation, was downloaded from the iGenomes site. Transcript assembly and differential expression analyses were performed using Cufflinks (v2.1.1). Assembly of novel transcripts was not allowed (-G), other parameters of Cufflinks followed the default setting. The summed RPKM (reads per kilobase per million mapped reads) of transcripts sharing each gene_id were calculated and exported by the Cuffdiff

program. In the RPKM data matrix, a minimal RPKM value of 0.1 was applied if gene expression was less than this minimal value. P-values were calculated using T-test.

RNA-seq data of WT and *Zfp281*KO ESCs and EpiLCs from our previous study (accession: GSE81042) were processed with the same settings.

## Gene Ontology (GO) analysis

Gene ontology analyses were performed using the DAVID gene ontology functional annotation tool (http://david.abcc.ncifcrf.gov/tools.jsp) with all NCBI Mus musculus genes as a reference list.

## Geneset enrichment analysis (GSEA)

GSEA (v2.1.0, available at http://www.broadinstitute.org/gsea) was used to determine whether the set of genes of interest was statistically enriched in *Zfp281*KO versus WT embryo RNA-seq data. The genesets used for this study were derived from the Reactome pathway database. The enrichment plot, normalized enrichment score (NES) and q-value (FDR) were indicated for each enrichment test.

## ChIP-seq analysis

ChIP was performed as previously described (*Lee et al., 2006*) using a Zfp281 antibody (Abcam, ab101318) in EpiSCs. ChIP-seq data of Zfp281 in ESCs were from our previous data set at GEO (accession: GSE81042). Massively parallel sequencing was performed with the Illumina HiSeq 2500 Sequencing System according to the manufacturer's protocol. All libraries were sequenced as 100 bp single-end reads. ChIP-seq raw data were processed as previously described (*Fidalgo et al., 2016*). Briefly, reads were aligned to the mouse genome (NCBI build 37, mm9) using the Bowtie (v1.0.0) program, with parameters -m 1. The aligned reads were sorted, and duplicated reads were removed using samtools (v0.1.19). BAM files were converted to a binary tiled file (tdf), and visualized using IGV (v2.3) software.

Zfp281 ChIP-seq peaks were determined using the MACS2 program (v2.0.10) with input ChIP-seq as the control. MACS2 parameters followed the default settings. Zfp281 ChIP-seq peaks were annotated using the annotatePeaks module in the HOMER program (v4.6) against the mm9 genome. Overlap of Zfp281 ChIP-seq peaks in ESCs and EpiSCs was determined by Bedtools (v2.18.1). The diffbind package (available from http://bioconductor.org/, v1.16.3) was used to determine the ChIP-seq intensities (by aligned reads per million total reads, RBM) of Zfp281 peaks.

Public ChIP-seq data were downloaded from GEO, and processed with the same settings. Colocalization of Zfp281 and other transcription and epigenetic factors (see accessions in *Figure 6— source data 1*) was performed with an in-house Python (v2.7.6) program as previously described (*Ding et al., 2015*). Briefly, ChIP-seq reads were downloaded and processed to determine a peak list by MACS2. A phi correlation coefficient was used to calculate the correlation between the peak lists of every two ChIP-seq data. Heatmap of correlations was shown with the TreeView program.

## Accession number

Zfp281 ChIP-seq data and *Zfp281*KO embryo RNA-seq data are available in GEO under accession GSE93044.

## Acknowledgements

We thank Dr. SY Kim and the Rodent Genetic Engineering Laboratory at NYU for the generation of tetraploid embryo chimeras; Drs. M Goll, S Nowotschin, X Shen, C Simon, P Soriano and T Wu for critical reading of the manuscript. This research was funded by grants from the National Institutes of Health (NIH) to JW (R01-GM095942 and R21-HD087722), the Empire State Stem Cell Fund through New York State Department of Health (NYSTEM) to JW (C028103, C028121). JW is a recipient of Irma T Hirschl and Weill-Caulier Trusts Career Scientist Award. Work in A-KH's lab is funded by NYSTEM (C029568) and the NIH (R01-DK084391 and P30-CA008748).

## Additional information

### Funding

| Funder | Grant reference number | Author |
|--------|------------------------|--------|
| National Institutes of Health | R01-GM095942 | Jianlong Wang |
| National Institutes of Health | R21-HD087722 | Jianlong Wang |
| New York State Department of Health | C028103 | Jianlong Wang |
| New York State Department of Health | C028121 | Jianlong Wang |
| National Institutes of Health | R01-DK084391 | Anna-Katerina Hadjantonakis |
| National Institutes of Health | P30-CA008748 | Anna-Katerina Hadjantonakis |
| New York State Department of Health | C029568 | Anna-Katerina Hadjantonakis |
| The Irma T. Hirschl/Monique Weill-Caulier Trust | Career Scientist Award | Jianlong Wang |

The funders had no role in study design, data collection and interpretation, or the decision to submit the work for publication.

### Author contributions

Xin Huang, Conceptualization, Resources, Data curation, Methodology, Writing—original draft, Writing—review and editing; Sophie Balmer, Conceptualization, Data curation, Methodology, Writing—original draft, Writing—review and editing; Fan Yang, Miguel Fidalgo, Dan Li, Diana Guallar, Methodology; Anna-Katerina Hadjantonakis, Jianlong Wang, Conceptualization, Writing—original draft, Project administration, Writing—review and editing

### Author ORCIDs

Xin Huang http://orcid.org/0000-0001-6778-8849
Sophie Balmer http://orcid.org/0000-0002-6561-552X
Miguel Fidalgo http://orcid.org/0000-0003-1134-2674
Anna-Katerina Hadjantonakis http://orcid.org/0000-0002-7580-5124
Jianlong Wang http://orcid.org/0000-0002-1317-6457

### Ethics

Animal experimentation: All mice used in this study were maintained in accordance with the guidelines of the Memorial Sloan Kettering Cancer Center (MSKCC) Institutional Animal Care and Use Committee (IACUC) under protocol number 03-12-017 (PI Hadjantonakis).

### Decision letter and Author response

Decision letter https://doi.org/10.7554/eLife.33333.049
Author response https://doi.org/10.7554/eLife.33333.050

## Additional files

### Supplementary files

• Transparent reporting form
DOI: https://doi.org/10.7554/eLife.33333.031

### Major datasets

The following dataset was generated:

| Author(s) | Year | Dataset title | Dataset URL | Database, license, and accessibility information |
|---|---|---|---|---|
| Huang X, Wang J | 2017 | Zfp281 is essential for epiblast maturation through transcriptional control of Nodal Signaling in the mouse embryo | https://www.ncbi.nlm. nih.gov/geo/query/acc. cgi?acc=GSE93044 | Publicly available atthe NCBI Gene Expression Omnibus (accession no: GSE93044) |

The following previously published datasets were used:

| Author(s) | Year | Dataset title | Dataset URL | Database, license, and accessibility information |
|---|---|---|---|---|
| Fidalgo M, Huang X, Wang J | 2016 | Zfp281 Coordinates Opposite Functions of Tet1 and Tet2 for Alternative Pluripotent States | https://www.ncbi.nlm. nih.gov/geo/query/acc. cgi?acc=GSE81045 | Publicly available at the NCBI Gene Expression Omnibus (accession no: GSE81045) |
| Hon G, Ren B | 2014 | 5mC Oxidation by Tet2 Modulates Enhancer Activity and Timing of Transcriptome Reprogramming during Differentiation | https://www.ncbi.nlm. nih.gov/geo/query/acc. cgi?acc=GSE48519 | Publicly available at the NCBI Gene Expression Omnibus (accession no: GSE48519) |
| Buecker C | 2014 | Reorganization of enhancer patterns in transition from naïve to primed pluripotency | https://www.ncbi.nlm. nih.gov/geo/query/acc. cgi?acc=GSE56098 | Publicly available at the NCBI Gene Expression Omnibus (accession no: GSE56098) |
| Fazzio TG, Chen PB | 2015 | Promoter-proximal R-loops regulate binding of chromatin regulators and pluripotency | https://www.ncbi.nlm. nih.gov/geo/query/acc. cgi?acc=GSE67580 | Publicly available at the NCBI Gene Expression Omnibus (accession no: GSE67580) |
| Young RA | 2008 | Connecting microRNA genes to the core transcriptional regulatory circuitry of embryonic stem cells | https://www.ncbi.nlm. nih.gov/geo/query/acc. cgi?acc=GSE11724 | Publicly available at the NCBI Gene Expression Omnibus (accession no: GSE11724) |
| Whyte W, Bilodeau S, Hoke H, Orlando DA, Frampton GM, Young RA | 2012 | Enhancer Decommissioning by LSD1 During Embryonic Stem Cell Differentiation | https://www.ncbi.nlm. nih.gov/geo/query/acc. cgi?acc=GSE27841 | Publicly available at the NCBI Gene Expression Omnibus (accession no: GSE27841) |

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
