## [Decision Letter]

[Editors’ note: a previous version of this study was rejected after peer review, but the authors submitted for reconsideration. The first decision letter after peer review is shown below.]

Thank you for submitting your work entitled "Zfp281 is essential for epiblast maturation through transcriptional control of Nodal Signaling in the mouse embryo" for consideration by *eLife*. Your article has been reviewed by three peer reviewers, and the evaluation has been overseen by a Reviewing Editor and a Senior Editor. The reviewers have opted to remain anonymous.

Our decision has been reached after consultation between the reviewers. Based on these discussions and the individual reviews below, we regret to inform you that your work will not be considered further for publication in *eLife*.

You will see that two of the reviewers are positive and we do not disagree with the positive comments of our fellow reviewers. However, we do not think that a description of a mouse gastrulation phenotype is sufficient for publication in a high profile journal today. Our particular concern with this paper is that both parts are incomplete and are being forced together to fit a particular interpretation which may be only partly correct or even completely incorrect. To summarise briefly: the authors are working on a gene of interest in embryonic stem cell biology and here they describe a requirement for proper axis formation and gastrulation. Based on altered expression patterns in the mutant embryos they attribute this phenotype to disruption of Nodal signalling. They then provide genome location data indicating that potentially Zfp281 might regulate Nodal directly. So far, so good. What they do not go on to provide is direct evidence that Nodal or Nodal signalling are perturbed either in the embryo prior to overt phenotype manifestation or in the in vitro ES cell system. Nor do they determine which effects are cell autonomous in the epiblast. This is central to their argument that Zfp281 drives pluripotency progression. The missing experiments are straightforward for investigators of this experience and expertise and in would be expected by any developmental or stem cell biology journal and general interest journal too.

Reviewer #1:

The transcription factor Zfp281 was recently shown to be a key regulator of primed pluripotency in vitro (Fidalgo et al., 2016). In this manuscript, the Wang group focuses on the role of Zfp281 in vivo, elegantly and comprehensively describing a critical requirement for Zfp281 during the earliest stages of mouse post-implantation development.

Between the pre-implantation and early post-implantation periods, Zfp281 expression resolves from a broader distribution within the blastocyst (all 3 lineages) to the egg cylinder epiblast. (There is pesky background staining with their antibody in the visceral endoderm in Figure 1.) The authors next derive mice from a previously published mutant Zfp281 mouse ES cell line (Fidalgo 2011 Stem Cells). Zfp281 homozygous embryos are morphologically distinct from their WT littermates by E6.5, with a thickened distal visceral endoderm and smaller size, and do not survive beyond embryonic day (E) 8.0. Thus, the authors performed comparative RNA-seq on whole wild-type and Zfp281-/- E6.5 embryos in an effort to identify the gene networks dysregulated by loss of Zfp281. The most significantly downregulated genes in Zfp281 mutants were prominent components of the Nodal signaling pathway (e.g., Nodal itself, Foxh1, Lefty1/2) as well as genes that display specific spatiotemporal expression domains in the epiblast such as the posterior primitive streak or visceral endoderm. Posterior primitive streak markers (e.g., Fgf8, T/Bra, Nodal, Eomes) showed a "radialized" expression pattern by whole-mount, that is, situated proximally in the epiblast abutting the extraembryonic ectoderm. Consistent with this finding, AVE markers were rarely expressed, or only expressed at low levels in the distal visceral endoderm, which fails to migrate anteriorly. Taken together these findings indicate a failure to establish the primary anterior-posterior axis, suggesting that Zfp281 mutant embryos are stuck at the precipice to axis and lineage specification.

To determine those genes directly bound and regulated by Zfp281, the authors turned to an in vitro model of naïve (ESC) and primed (epiblast-like cells, EpiLC) pluripotency and performed Zfp281 ChIP-seq. Interestingly, the promoters of lineage-specific genes, which are typically expressed at higher levels in primed cells (EpiLC and EpiSC) than ESC (Figure 5), were bound by Zfp281 with greater enrichment in ESC than in EpiLC (Figure 5). The authors interpret these findings to mean that Zfp281 regulates lineage-specific genes during epiblast "maturation." They next combined their Zfp281 ChIP-seq data with publically available binding data sets for epigenetic regulators and transcription factors. Zfp281, Ep400 and the PRC2 components Ezh2 and Suz12 have the most similar binding patterns, with Zfp281/Ep400 and Suz12 co-binding observed at bivalent genes marked by co-enrichment of active H3K4me3 and repressive H3K27me3. During the transition from naïve to primed pluripotency, the authors propose that there is a reduced association between Zfp281 and the PRC2 complex at bivalent promoters, resulting in decreased H3K27me3 and transcriptional activation of lineage-specific genes. Lastly, the authors revisit the role of Zfp281 and transcriptional activation of the Nodal locus and determine that Oct4 and P300 collaborate with Zfp281 to activate Nodal and is components Foxh1 and Lefty2 during epiblast "maturation." Their findings support a model of enhancer switching from HBE to ASE as the epiblast exits pluripotency.

This reviewer finds the present work to be both comprehensive and insightful, certainly meriting publication in *eLife* and highlighting the importance of in vivo studies to more fully understand the role of Zfp281 previously described only in vitro.

Reviewer #2:

In this manuscript, Huang et al. characterise the function of the zinc finger protein, Zfp281, in early mouse development during exit from pluripotency, axis specification and lineage priming. Zfp281 was previously identified from ES cell screening as a potential candidate for facilitating differentiation in vitro. The authors provide a detailed analysis of the expression profile of Zfp281 in mouse embryos at significant stages around the time of implantation and gastrulation, including quantification of levels of fluorescence. They show enriched expression in the epiblast from E4.25 until early postimplantation. The knockout phenotype begins very soon after implantation, with a thickening of the visceral endoderm and reduced proliferation of the epiblast. Thereafter, the mutant embryo lacks anterior-posterior patterning with no evidence of gastrulation, explaining the absence of homozygous null embryos from E8.5. Using gene expression profiling of null versus wild type embryos the authors identify Nodal as the most affected signalling pathway in mutant embryos, with Nodal itself and its downstream targets being underrepresented in null embryos by E6.5. As expected, the perturbation of the Nodal pathway affects the anterior-posterior patterning of mutant embryos and prevents specification of the primitive streak. This was manifested by mis-regulation of Nanog, T, Eomes, FGF8, Axin2, Otx2 and Sox2, even though their total expression levels were largely unaffected. Many of the known anterior markers were lost or disrupted in mutant embryos. This explains the encroachment of primitive streak genes to the anterior epiblast in mutant embryos.

To address the mechanism of Zfp281 activity during exit from naïve pluripotency the authors used EpiLCs generated after 2 days culture of ES cells in EpiSC conditions. They used ChIP-seq to show that Zfp2981 is enriched at regulatory regions of multiple genes with both conservation and divergence between ES cells and EpiLCs, implying a dynamic role during differentiation. In particular, Zfp281 bound to early lineage genes in ES cells. Interestingly, only the enhancers and not the promoters of FGF5 and 8 were bound by Zfp281, implying both promoter dependent and independent mechanisms for its action. Zfp281 associates with specific sets of epigenetic regulators, which differ between ES cells and EpiSCs. There was also reduced chromatin occupancy at bivalent promoters for T and Eomes in mutant compared with wild type ES cells. The authors suggest that Zfp281 is actively involved in reduction of H2K27me3 during exit from pluripotency. Nodal enhancer activity was explored in ES cells and EpiSCs and confirmed to be dynamic during the transition from naïve to primed pluripotency in vitro. Direct evidence for the mechanism of action of Nodal during lineage priming in vivo was obtained by performing ChIP experiments on wild type E6.5 embryos. Zfp281 binding was observed in regulatory regions of Nodal and some of its targets affected in mutant embryos.

This study is beautifully planned, executed and presented with sufficient novel and interesting findings to enhance understanding of the mechanism of exit from pluripotency and lineage priming and the role of Nodal signalling during this process in vivo and in vitro.

Reviewer #3:

In this paper Huang et al. first describe the early lethality in mouse embryos deficient for Zfp281 and then apply ChIP-seq in ES cells and EpiSCs. Previous studies from Jianlong Wang have implicated Zfp281 in regulation of both ES cell self-renewal and early differentiation. Given a strong in vitro effect on ES cell differentiation there is interest in elucidating the stage, character and causality of the in vivo phenotype. In the present study the authors infer a relationship with Nodal signalling in early embryogenesis and suggest that Zfp281 lies directly upstream of Nodal itself. This would be very interesting, if proven. The latter section of the paper, comprising genome-wide analyses in ES cells and EpiSCs, is only loosely connected to the embryo phenotype and doesnot provide mechanistic data. Overall, neither component in the study is taken through to completion and the whole is rather less than the sum of the parts. In the end we are still little wiser as to the functional role and mode of action of Zfp281 in either embryonic development or ES cell differentiation.

The first section of the paper reports a gastrulation stage phenotype in Zfp281 null embryos and notes common features with Nodal mutants. This work is well done and the results are clearly presented and described. However, the study is incomplete. If the phenotype is attributed to dysregulation of the Nodal pathway we need to know whether expression of Nodal components is indeed disrupted prior to the overt phenotype, i.e. at E5.5. More generally, phenotypic analysis of the peri-implantation period is critical for the central argument of the paper that Zfp281 drives progression from naïve pluripotency. Are genes that are up-regulated early in transition, such as Otx2 and Pou3f1, expressed correctly or not in mutant embryos at E5.5? Conversely, it is necessary to determine whether the epiblast defects are indeed cell-autonomous, as the authors argue, or are entirely secondary to failure of the DVE and subsequent absence of AVE. Tetraploid blastocyst complementation is well-established as the method to address this issue. Alternatively the authors could create an epiblast-specific deletion using Sox2::Cre or other suitable driver. These experiments will take time, but frankly without them the present description does not allow any conclusions as to the primary defect and cell autonomous versus non-autonomous action. The RNA-seq analyses appear to have been carried out on whole embryos. Validation by ISH or IF to confirm tissue/region, at least for up-regulated genes such as AFP, would be expected.

The second part of the paper features molecular analyses in vitro. The authors present correlations from ChIP-seq analyses but they do not test directly any of the inferences. The key advantage of in vitro stem cell systems is that they enable not just high throughput data collection but also direct and rigorous examination of hypothesised regulatory interactions. Unfortunately there is no such test of what I take to be the main hypothesis, that Zfp281 directly regulates Nodal expression and Nodal signalling. Is Smad2/3 phosphorylation affected in the mutants? To what extent is the Zfp281 phenotype recapitulated by inhibition and/or genetic ablation of nodal signalling. The authors report that gene expression is perturbed in EpiLCs but the protocol for generating EpiLCs involves addition of activin, so should not be affected by diminished nodal expression.

Previous reports from the Wang lab have asserted that Zfp281 acts via Nanog and NuRD and most recently Tets. The current manuscript claims further modes of action; interaction with Tip60-Ep400m, and direct regulation of Nodal and nodal pathway components. But how are these different mechanisms integrated and what is their relative significance. The study does not provide any coherent synthesis. Simply saying Zfp281 is a master regulator is not really sufficient. We are left with insufficient clarity as to how and when Zfp281 is required in vitro or in vivo.

---

## [Author Response]

[Editors’ note: the author responses to the first round of peer review follow.]

Reviewer #1:[…] This reviewer finds the present work to be both comprehensive and insightful, certainly meriting publication in eLife and highlighting the importance of in vivo studies to more fully understand the role of Zfp281 previously described only in vitro.

We thank this reviewer for his/her comments which are both supportive of publication of our manuscript in *eLife*, as well as recognizing the significance of this work, and the importance of undertaking in vivoand in vitrostudies to fully understand the function of Zfp281.

Reviewer #2:[…] This study is beautifully planned, executed and presented with sufficient novel and interesting findings to enhance understanding of the mechanism of exit from pluripotency and lineage priming and the role of Nodal signalling during this process in vivo and in vitro.

We thank this reviewer for his/her positive and supportive comments on the planning and execution of our study, and the significance of our findings.

Reviewer #3:In this paper Huang et al. first describe the early lethality in mouse embryos deficient for Zfp281 and then apply ChIP-seq in ES cells and EpiSCs. Previous studies from Jianlong Wang have implicated Zfp281 in regulation of both ES cell self-renewal and early differentiation. Given a strong in vitro effect on ES cell differentiation there is interest in elucidating the stage, character and causality of the in vivo phenotype. In the present study the authors infer a relationship with Nodal signalling in early embryogenesis and suggest that Zfp281 lies directly upstream of Nodal itself. This would be very interesting, if proven.

We appreciate the reviewer's interest in our work. However, we want to clarify that our data, for example our in vitroand in vivoChIP-qPCR experiments presented in Figure 7, support the direct binding of Zfp281 on the *Nodal* promoter during epiblast maturation. Ours is the first study to demonstrate direct in vivobinding at the *Nodal* locus. We would therefore argue that we do not *per se* infer a role for a relationship between Zfp281 and Nodal signaling. Our suggestion of Zfp281 lying upstream of Nodal comes from direct experimental evidence (binding of Zfp281 to *Nodal cis-*regulatory elements in vitroin pluripotent stem cell models, and in vivoin embryos) provided in our manuscript.

We would also like to point out the two other reviewers of our manuscript also noticed these data, and commented that "Their findings support a model of enhancer switching from HBE to ASE as the epiblast exits pluripotency" (Reviewer 1) and "Zfp281 binding was observed in regulatory regions of Nodal and some of its targets affected in mutant embryos." (Reviewer 2).

The latter section of the paper, comprising genome-wide analyses in ES cells and EpiSCs, is only loosely connected to the embryo phenotype and doesnot provide mechanistic data. Overall, neither component in the study is taken through to completion and the whole is rather less than the sum of the parts. In the end we are still little wiser as to the functional role and mode of action of Zfp281 in either embryonic development or ES cell differentiation.

We respectfully disagree with the reviewer's comment that the in vitro and embryo data are "only loosely connected", as both sets of experiments cross-reference one another and have cross-confirmatory findings. These correlative in vitro and in vivo studies together forged the link to Nodal signaling, which was confirmed by performing technically challenging ChIP-qPCR experiments on embryos. We would also like to point out that our present study is one of the first to demonstrate direct binding of a transcription factor to cis-regulatory elements in vivo in mouse embryos.

We believe that the reviewer's comment that our genome-wide analyses in ES cells and EpiSCs do not provide mechanistic data explaining the embryo phenotype was unfair and unjustified. The in vitro ESC-to-EpiSC differentiation system coupled with genome-wide analyses have been widely applied in providing mechanistic understanding of transcriptional and epigenetic regulation underlying naïve-to-primed pluripotency transition (work from the Surani, Wysocka, Smith labs, as well as our own previous work).

Perhaps the most pertinent example is a study from Austin Smith's laboratory (Cambridge, UK), recently published in the journal Stem Cell Reports (Mulas et al., 2017), in which the authors utilized an in vitro ESC-to-EpiSC model and uncovered a role for Nodal in securing pluripotency upon ESC progression from the ground state. Importantly, in their Discussion, Mulas et al. concluded that "Our findings point to a pivotal role for NODAL signaling in establishing formative pluripotency, in keeping with observations of a requirement for continuous NODAL activity to sustain pluripotency in the early postimplantation epiblast".

Similarly, we have also provided a careful discussion in our manuscript for how our in vitro findings may offer mechanistic insights into molecular control underlying the early postimplantation epiblast, and transitions between pluripotent states. More importantly, our data supports a model whereby Zfp281 directly binds the Nodal promoter during epiblast maturation (as shown by both our in vitro and in vivo ChIP-qPCR data in Figure 7), and that Nodal expression is affected by loss of Zfp281 in the embryo. We also show the direct regulation by Zfp281 of genes that need to be activated for the epiblast to properly mature.

We have added more data, as detailed below, to strengthen the statements made and hypotheses arising from our work.

The first section of the paper reports a gastrulation stage phenotype in Zfp281 null embryos and notes common features with Nodal mutants. This work is well done and the results are clearly presented and described.However, the study is incomplete. If the phenotype is attributed to dysregulation of the Nodal pathway we need to know whether expression of Nodal components is indeed disrupted prior to the overt phenotype, i.e. at E5.5. More generally, phenotypic analysis of the peri-implantation period is critical for the central argument of the paper that Zfp281 drives progression from naïve pluripotency. Are genes that are up-regulated early in transition, such as Otx2 and Pou3f1, expressed correctly or not in mutant embryos at E5.5?

We agree with the reviewer that these data on earlier embryos would strengthen our manuscript and support our conclusions. We have now included ISH/immunostaining for the suggested markers at earlier stages: additional IFs are added in Figure 3—figure supplement 2 to strengthen our conclusions that markers of the formative state (Otx2 and Oct6/Pou3f1) are downregulated in *Zfp281*KO embryos before migration of the AVE/DVE and after the embryo's implantation (E5.5). To our knowledge, Oct6 protein distribution at this stage has never been published (only WISH data were published at early post-implantation (Zhu et al., 2014) or by RNA-seq of E5.5 embryos (Kalkan et al., 2017) and as noted in Figure 3—figure supplement 2, seems to be quite mosaic in embryos.

We also added WISH for Nodal at E5.5 in Figure 4—figure supplement 2, showing that its expression is already downregulated in E5.5 *Zfp281*KO embryos when compared to WT.

Conversely, it is necessary to determine whether the epiblast defects are indeed cell-autonomous, as the authors argue, or are entirely secondary to failure of the DVE and subsequent absence of AVE. Tetraploid blastocyst complementation is well-established as the method to address this issue. Alternatively the authors could create an epiblast-specific deletion using Sox2::Cre or other suitable driver. These experiments will take time, but frankly without them the present description does not allow any conclusions as to the primary defect and cell autonomous versus non-autonomous action.

Unfortunately, at present there is no conditional allele available for *Zfp281* so the epiblastspecific ablation is not currently possible. The mouse *Zfp281* locus has a very unusual structure that does not make it easily amenable to conditional mutagenesis. As an alternative approach to assessing the epiblast-specific function of Zfp281, and as suggested by the reviewer, we have generated 4n (tetraploid) wild-type embryo <-> *Zfp281* mutant ESC chimeras. In such a tetraploid complementation experiment, the 4n tissues (extra-embryonic Primitive Endoderm and Trophectoderm derivatives) are wild-type, whereas all epiblast derivatives are *Zfp281* mutant. This, in principle, is an analogous experiment to an epiblast-specific deletion of a conditional mutant allele.

These 4n blastocyst complementation experiments have allowed us to clarify the epiblast specificity of Zfp281, and in doing so extend and support the immunofluorescent localization data presented in our original submission.

In accordance with our model, 4n wild-type embryo <-> *Zfp281* mutant ESC chimeras recapitulate the *Zfp281* null mutant phenotype.

· 4n Zfp281KO ESC chimeric embryos exhibit thickened VE layer similarly to Zfp281KO embryos (added in Figure 1—figure supplement 3).

· As in *Zfp281*KO embryos, some pluripotency associated markers are also downregulated in *Zfp281*KO ESC chimeras at E5.75-6.0 (added in Figure 3—figure supplement 2).

· We note that *Zfp281*KO ESC chimeras exhibit a failure to establish the anterior-posterior axis (added in Figure 4—figure supplement 2).

· 4n Zfp281KO ESC chimeric embryos exhibit the same pattern of expression of Nanog and T as Zfp281KO embryos at E7.25 (Figure 3—figure supplement 3).

· Furthermore, we go on to show that 4n wild-type embryo <-> cDNA rescued Zfp281 mutant ESC chimeras exhibit a partial rescue of the *Zfp281* mutant phenotype.

In conclusion, these new 4n (tetraploid complementation) chimera data support our proposed model for Zfp281 exhibiting a cell autonomous function within the epiblast, which was originally based on our demonstration of Zfp281 being exclusively expressed within the epiblast at the time of implantation (and thereafter).

Importantly, these 4n chimera data (and neither an epiblast-specific ablation using a conditional allele) will not allow us to answer the question as to whether the defects we observe are due to a failure of DVE/AVE specification and/or migration. We would argue that experiments addressing this question are beyond the scope of the present study.

The RNA-seq analyses appear to have been carried out on whole embryos. Validation by ISH or IF to confirm tissue/region, at least for up-regulated genes such as AFP, would be expected.

We have already provided validation of RNA-seq data by ISH or IF for a number of downregulated genes in our manuscript. For up-regulated genes such as Afp as suggested, however, a working antibody for AFP IF is lacking. We nonetheless performed additional RT-qPCR on embryos and confirmed their up-regulation for a number of genes (data are added in Figure 2) and have added more explanations and discussion (paragraph two) in our revised manuscript.

The second part of the paper features molecular analyses in vitro. The authors present correlations from ChIP-seq analyses but they do not test directly any of the inferences. The key advantage of in vitro stem cell systems is that they enable not just high throughput data collection but also direct and rigorous examination of hypothesised regulatory interactions. Unfortunately there is no such test of what I take to be the main hypothesis, that Zfp281 directly regulates Nodal expression and Nodal signalling. Is Smad2/3 phosphorylation affected in the mutants? To what extent is the Zfp281 phenotype recapitulated by inhibition and/or genetic ablation of nodal signalling. The authors report that gene expression is perturbed in EpiLCs but the protocol for generating EpiLCs involves addition of activin, so should not be affected by diminished nodal expression.

In accordance with the reviewer’s comments, the following additional experiments were performed to dissect the functional connection between Zfp281 and Nodal signalingin vitroin ESCs and differentiation.

Firstly, we determined the level of phospho(p-)Smad2 in WT and *Zfp281*KO ESCs. As shown in our new Figure 7-Smad2 intensity in *Zfp281*KO ESCs was significantly diminished compared to that observed in WT ESCs.

We previously demonstrated that endogenous Zfp281 is required for survival of EpiSCs or longterm culture of EpiLCs in primed conditions containing Activin (Fidalgo et al., 2016), suggesting that Zfp281 is important for the integrity of Nodal signaling. In this regard, we examined levels of p-Smad2 in EpiSCs with Zfp281 knockdown. As shown in the new Figure 7—figure supplement 1, knockdown of Zfp281 didn’t change p-Smad2 activity in EpiSCs, probably because of the Activin in culture constitutively activates p-Smad2. However, we did observe that the expression of Lefty (using an antibody detecting both Lefty1/2) significantly decreased. This is in accordance with our ChIP-qPCR results that Zfp281 binds to *Lefty2* enhancer in in vitro ESCs/EpiSCs and in vivo E6.5 embryos (Figure 7).

Furthermore, we treated ESCs and EpiSCs with ALK receptor inhibitor (ALKi) specifically diminishing Smad2/Nodal signaling. As shown in the new Figure 7—figure supplement 1 (copied below), ALKi treatment significantly reduces pSmad2 and Lefty expression.

In summary, our new data further demonstrated the functional role of Zfp281 in regulating Nodal signaling, particularly in EipSCs with Nodal signaling activation, Zfp281 is required to transcriptionally activate Nodal and downstream target *Lefty* in vitro (Figure 5—figure supplement 1, Figure 7).

To further understand the functional significance of Zfp281 in controlling the Nodal signaling pathway during embryonic development, we employed the following four in vitrodifferentiation models that relate to Activin/Nodal signaling: (1) ESCs to EpiLCs; (2) ESCs to primitive streak (PS)-like cells; (3) ESCs to definitive endoderm (DE)-like cells; and (5) ESCs to PGCLCs.

1) With Fgf2 and Activin, ESCs can differentiate into EpiLCs representing a subsequent stage of pluripotency, and we have previously shown that *Zfp281*KO is not compatible with the longterm culture of EpiLCs (Fidalgo et al., 2016).

2) It is reported that Activin and GSK3β inhibitor (CHIR) promote differentiation of ESCs to primitive-streak (PS)-like cells (Tsakiridis et al., 2014). We therefore tested the role of Zfp281 in this differentiation. As shown in new Figure 7—figure supplement 2, we treated ESCs with Activin/CHIR, and with ALKi or Zfp281 shRNAs. Morphology of Zfp281 knockdown cells indicated a strong phenotype of differentiation resistance, which resembles that of ALKi treatment. Dome-shaped ESC-like colonies persisted in Zfp281 KD cells after Activin/CHIR treatment; a striking difference compared to control cells. RT-qPCR at different time points after treatment also indicated that the Zfp281 KD cells reproduced Nodal signaling inhibition by ALKi in terms of the expression of the PS marker gene *T* and the Nodal target gene *Lefty2* (expressed in PS).

3) Activin/Nodal activity is required forin vitro differentiation of ESCs to definitive endoderm (DE)-like cells. According to a protocol developed by Gouon-Evans et al. (Gouon-Evans et al., 2006), DE differentiation starts within 2-days of embryonic body (EB) formation (Day2) in serumfree medium, then followed by additional 2~3 days (Day4~Day5) of Activin/Nodal activation. We first confirmed that WT ESCs (J1, CJ7) can form DE-like cells in our hands, indicated by robust activation of DE marker *Foxa2*. We then performed DE differentiation of *Zfp281*KO ESCs. At Day 2 of EB formation, similar percentages of live cells were obtained, indicating that Zfp281 is dispensable in forming EBs as we previously reported (Fidalgo et al., 2011). However, significant cell death was observed upon DE differentiation (Day4~Day5) (Huang et al., manuscript under preparation). Since DE is a descendant lineage of epiblast, these data suggest that the failure of *Zfp281*KO ESCs to differentiate into DE may be due to an overall requirement for Zfp281 in epiblast development, which is consistent with the failure of *Zfp281*KO ESC in the ESC-to-EpiSC transition.

4) A recent study from Austin Smith’s lab (Cambridge, UK) has reported a role for Nodal signaling in differentiation of ESCs to Primordial Germ Cell Like Cell (PGCLC) (Mulas et al., 2017). Following the protocol of PGCLC differentiation from Mitinori Saitou’s lab (Hayashi et al., 2011), we firstly adapted WT and *Zfp281*KO ESCs in 2iL (2i+LIF) conditions for 2 passages (>10 days), followed by 2 days of EpiLC differentiation, and 4 days in conditions promoting PGCLC formation. Both WT and *Zfp281*KO cells can form PGCLCs. By assaying the mRNA expression of PGC markers (*Nanos3, Stella, Tfap2c, Nanog, Prdm1, Prdm14*), we found that these genes generally followed a similar trend in PGCLC differentiation. Moreover, we observed that many of them exhibited higher levels of expression in *Zfp281*KO cells compared to WT cells, in the initial 2iL naive ESC state and the final PGCLC state (Author response image 1).

These data differ from Smith’s results in which inhibition of Nodal signaling led to a reduction in the expression of PGC marker genes. However, since Activin is not used in the standard PGCLC differentiation protocol, it is still not clear how Nodal signaling can contribute to PGCLC formation. Furthermore, Mulas et al. indicated Activin treatment can rescue the Nodal KO phenotype in PGCLC formation, and speculated that expression of Gdf3, a Vg-1 homolog, may elicit a Nodal-like response (Mulas et al., 2017). In our experiments, we find PGC markers to also be expressed in naive ESCs, and we previously showed these naive markers (Nanog, Stella, Prdm14) are repressed by Zfp281 in 2iL ESCs (Fidalgo et al., 2016). These observations may, at least in part, explain the higher levels of PGC marker genes observed in *Zfp281*KO cells.

Thus, in our attempts to answer the reviewer’s question “To what extent is the Zfp281 phenotype recapitulated by inhibition and/or genetic ablation of nodal signaling”, we applied multiple in vitro differentiation protocols (EpiLC, PS, DE) from WT/*Zfp281*KO ESCs especially in contexts in which Activin/Nodal signaling is essential for the respective differentiation. We also showed *Zfp281* KD can reproduce Nodal signaling inhibition in terms of cell morphology change upon differentiation and expression of Nodal signaling targets. However, we did not directly test the outcome of *Nodal* KO because the function/requirement of/for Nodal has been well-studied in many of these contexts, and is outside the scope of our present study. Furthermore, even with *Nodal* KO cells, it has been reported that Nodal signaling is unaffected upon Activin treatment with respect to the expression of pathway target genes, for example *Lefty2* (Mulas et al., 2017).

In our revised manuscript, we provide these additional in vitro ES cell differentiation data to support our hypothesis that Zfp281 is required for maintaining the integrity of Nodal signaling in in vitro ESC differentiation models. For DE differentiation, it is still unclear why *Zfp281*KO cells fail to differentiate and instead exhibit massive cell death/apoptosis. For the ESC-to-PGCLC differentiation paradigm, we believe that the function of Nodal signaling remains ill-defined, and thus we decided not to include our DE and PGCLC data in the revised manuscript.

Previous reports from the Wang lab have asserted that Zfp281 acts via Nanog and NuRD and most recently Tets. The current manuscript claims further modes of action; interaction with Tip60-Ep400m, and direct regulation of Nodal and nodal pathway components. But how are these different mechanisms integrated and what is their relative significance. The study does not provide any coherent synthesis. Simply saying Zfp281 is a master regulator is not really sufficient. We are left with insufficient clarity as to how and when Zfp281 is required in vitro or in vivo.

In this study, we focus on the function of Zfp281 within the epiblast of the early post-implantation mouse embryo, and comparablein vitroexit from pluripotency and concomitant lineage gene specification. In this way, the present study is complementary to, and indeed significantly extends, our previous knowledge on Zfp281 in pluripotency (Fidalgo et al., 2012; Fidalgo et al., 2016; Fidalgo et al., 2011). In the present study we uncover two regulatory machineries in which Zfp281 has a central and critical role: Zfp281/PRC2/Ep400 on Class I (bivalent) genes, and Zfp281/PRC2/Ep400 on Class II (pluripotency) genes (Figure 6). We have shown Zfp281 is a repressor for pluripotency-associated genes (through NuRD) in ESCs (Fidalgo et al., 2012), we therefore focused on the role of Zfp281 on bivalent/lineage specific genes in this work. Specifically, as we state in our manuscript: "The lack of association between Zfp281 and NuRD in EpiSCs may also explain why pluripotency genes including *Sox2* and Nanog are not regulated by Zfp281 in the epiblast (Figure 3), which is in contrast with NuRD-mediated Nanog repression in ESCs due to their physical association (Fidalgo et al. 2012)."

In our (E6.5) embryo qPCR (Figure 2) and immunofluorescent staining (Figure 3) data, we report no difference in Nanog expression between WT and *Zfp281*KO embryos, therefore we did not go on to further investigate Nanog as a target of Zfp281 at post-implantation stages. For the TET proteins, previously we showed that Zfp281 represses *Tet2* expression, but has no effect on *Tet1* (Fidalgo et al., 2016). As shown in Author response image 2, qPCR of E6.5 embryos indicated Tet1 is the highest expressed TET gene at this stage, while Tet2 exhibits highly reduced expression. We confirmed that Tet1 expression is not affected in *Zfp281*KO embryos, whereas Tet2 is appreciably upregulated. Moreover, Zfp281 physically interacts with Tet1 (but not Tet2), suggesting Zfp281/Tet1 may have similar function on their target genes (Fidalgo et al., 2016). Previously, Anjana Rao’s group (La Jolla, USA) reported that Tet1 is required for the demethylation of the *Lefty1* promoter in ESCs (Koh et al., 2011). More recently, a study from Guoliang Xu’s group (Shanghai, China) suggested that TET proteins are essential for the demethylation of *Lefty* promoters in vivo before the onset of gastrulation (Dai et al., 2016). Both these studies are supportive of our data reporting Zfp281 as an upstream regulator of the Nodal signaling pathway. The fact that Zfp281 is a key transcription factor that can not only coordinate the opposing functions of Tet1 and Tet2 during the naive-to-primed transition in vitro (Fidalgo et al., 2016), but also directly bind and control *Nodal* enhancer elements justifies referring to it as a master regulator.

**Author response image 2. respfig2:** 

Thus, we have provided explanations/coherent synthesis on these different mechanisms and their relative significance. In the revised manuscript, we have placed increased emphasis on the published work from other groups as well as ourselves, further clarifying a central and comprehensive role for Zfp281 in embryonic development (last paragraph of Discussion).

**References Cited:**

Gouon-Evans, V., Boussemart, L., Gadue, P., Nierhoff, D., Koehler, C.I., Kubo, A., Shafritz, D.A., and Keller, G. (2006). BMP-4 is required for hepatic specification of mouse embryonic stem cell-derived definitive endoderm. Nat Biotechnol 24, 1402-1411.

Kalkan, T., Olova, N., Roode, M., Mulas, C., Lee, H.J., Nett, I., Marks, H., Walker, R., Stunnenberg, H.G., Lilley, K.S., et al. (2017). Tracking the embryonic stem cell transition from ground state pluripotency. Development.

Zhu, Q., Song, L., Peng, G., Sun, N., Chen, J., Zhang, T., Sheng, N., Tang, W., Qian, C., Qiao, Y., et al. (2014). The transcription factor Pou3f1 promotes neural fate commitment via activation of neural lineage genes and inhibition of external signaling pathways. *eLife* 3.